# Dogs in Lithuania from the 12th to 18th C AD: Diet and Health According to Stable Isotope, Zooarchaeological, and Historical Data

**DOI:** 10.3390/ani14071023

**Published:** 2024-03-27

**Authors:** Giedrė Piličiauskienė, Raminta Skipitytė, Viktorija Micelicaitė, Povilas Blaževičius

**Affiliations:** 1Department of Archaeology, Vilnius University, Universiteto St. 7, 01513 Vilnius, Lithuania; raminta.skipityte@ftmc.lt (R.S.); viktorijamicelicaite@gmail.com (V.M.); povilas.blazevicius@gmail.com (P.B.); 2Isotopic Research Laboratory of Center for Physical Sciences and Technology, Savanorių Ave. 231, 02300 Vilnius, Lithuania

**Keywords:** dog, nutrition, health, size, morphotype, δ^13^C and δ^15^N analysis, Bayesian modelling, Middle Ages, early modern period, eastern Baltic

## Abstract

**Simple Summary:**

In this study, we discuss the dietary characteristics of different sizes and types of dogs (*n* = 75) from sites relating to different social strata and time periods in Lithuania from the 12th to the 18th C AD. Results demonstrate that the size, type, diet, and health of canines from different time periods and sociocultural environments varied, and elite dogs had different nutrition values to urban canines. Overall, carbon isotopic signals indicate that dogs’ diets were based on C_3_ plant environment foods (cereals and animals), while freshwater fish was more important for some individuals in coastal Klaipėda/Memelburg Castle. In the Middle Ages, the consumption of plant-based foods was likely higher compared to the early modern period, but this varied according to the particular individual. Our study also revealed that the diet was not related to the individual’s size. Compared to pigs, dogs had a higher intake of animal foods in their diet. In general, the nutrition of the studied canines was similar to that of the rural human population of the same period.

**Abstract:**

This article presents the results of research that focused on the nutrition and related health issues of medieval and early modern dogs found in the territory of present-day Lithuania. In this study, we present bone collagen carbon (δ^13^C) and nitrogen (δ^15^N) isotope ratios for seventy-five dogs recovered from seven sites which were dated back to the between the 12th and 18th C AD. In addition, by studying the remains of almost 200 dogs, we were able to estimate changes in the sizes and morphotypes of canines across over 600 years. On the basis of stable isotope and historical data, as well as the osteometric analysis, we discuss the dietary patterns of different sizes and types of dogs from the sites related to different social strata and time periods. The results of our study demonstrate that the size, type, diet, and health of canines from different time periods and sociocultural environments varied. Overall, carbon isotopic signals indicate that dogs’ diets were based on C_3_ plant environment foods (cereals and animals), while freshwater fish was more important for some individuals in coastal Klaipėda/Memelburg Castle. The stable isotope analysis supported the historical records, indicating that cereals were highly important in the diet of elite dogs. Meanwhile, urban dogs had a different nutrition. In the Middle Ages, the consumption of plant-based foods was likely higher compared to the early modern period. Our study also revealed that the diets of dogs did not correlate with individual size. Compared to pigs, dogs had a higher intake of animal foods in their diet. In general, the nutrition of the studied canines was similar to that of the rural human population of the same period.

## 1. Introduction

Dogs were undoubtedly the first animals domesticated by humans [1], and have been their closest companions for at least 14,000 years. Dogs are animals that, at least in the present-day Western world, humans call friends. Numerous studies have focused on the different aspects of the history and daily lives of canines, and their results are of great interest to the general public. Many studies have focused on the morphology, type, domestication, origin of breed, and nutrition of dogs in different regions and periods [2,3,4,5,6,7]. The latter topic may also be of great importance in human research, as dogs living within proximity of humans usually feed on waste from the tables or kitchens of those humans. Thus, some researchers have suggested that the canine diet may also reflect human nutrition [8,9]. However, such analogies are not fully supported by other studies [10,11,12,13].

The earliest undisputed remains of dogs in the territory of modern Lithuania were found in early Subneolithic (5000–2900 cal BC) sites, dating back to the early 4th millennium BC. A dog or a wolf tooth pendant was also found in the double grave of children, No. 5 in the Donkalnis cemetery, dated back to 6072–5920 cal BC [13]. Stone Age dogs in Lithuania were rather small, standing at approximately 40 cm tall. Although scarce, their remains were typically found in Stone Age sites containing faunal remains. However, canine remains were almost absent in Bronze Age (1800–500 cal BC) and Iron Age (400 cal BC-1200 cal AD) settlements. The earliest dog burials were recorded in Lithuania, dating to around 10th–11th C AD. Animals were buried in graves together with humans or horses. Hunting dogs were also cremated together with humans of various social strata. The end of the custom of burying dogs and horses is attributed to the Christianization of Lithuania in 1387, although some graves are dated as early as the beginning of the 15th C AD [14].

The abundant zooarchaeological material from medieval and early modern sites has also provided numerous dog remains. Although canine remains among the faunal assemblages of these periods represent up to 1–2% of the NISP, research has yielded quite detailed information regarding the dogs that lived in Lithuania in 12th–18th C AD. In addition, historical records about canines, their use, nutrition, care, and even prices already appeared in this period [14,15,16,17]. Thus, in general, quite a lot is known about the morphology, function, and types of dogs bred in this period.

However, information on dog nutrition is much scarcer. To date, only a few stable isotope, mainly carbon and nitrogen, studies have been carried out on dogs from present-day Lithuania. Five individuals from the Šventoji Subneolithic sites in coastal Lithuania have been studied. Their collagen carbon (from −23.8‰ to −19.2‰) and nitrogen (from 12.8‰ to 13.9‰) isotopic signatures differed from one another, indicating that canines, to some extent, were fed or ate freshwater fish, and probably also had access to marine foods. In general, the nutrition of Šventoji canines was very similar to the diet of the local Subneolithic humans. Only one dog from Šventoji had a strong marine stable isotope signal; however, the animal most likely originated from a medieval–early modern village [18,19]. High variability in dogs’ stable isotope values has also been documented for the Zvejnieki Mesolithic–Subneolithic cemetery in northern Latvia [10].

Analyses have also been performed on one dog from the Late Bronze Age (1100–500 cal BC) Turlojiškės (SW Lithuania) site, and one individual from the Roman Iron Age (1–400 cal BC) Eketė site (W Lithuania). Both animals had very similar isotopic signals, with δ^13^C of −21.6‰ and −21.8‰, and δ^15^N of 9.7‰ and 9.5‰, respectively [18,20]. These values demonstrate that the major dietary input was plant-based food, with the addition of some meat. A few more previously analyzed individuals from the early modern period [21,22] were reanalyzed in the current study.

The diet of Stone or Iron Age dogs in Lithuania can only be learned from stable isotope studies, although we can still make assumptions about the diet of prehistoric canines on the basis of the human economy and the results of studies on the human diet at the time. Meanwhile, historical records already provide information on the diet of medieval and post-medieval dogs. However, the latter only provide information on the nutrition of elite canines, mainly hunting dogs. For example, recommendations on how and what to feed game dogs in the early 17th century can be found in Jan Ostroróg’s work, published in Poland in 1618 [23], as well as in account books and other documents of nobles and the king [17,24]. The diet of urban canines can usually be determined only from carbon and nitrogen stable isotope analyses. Therefore, we combined stable isotope data and historical records on the diet of dogs, attempting to fill the gap in the knowledge of this topic.

In this paper, we present the results of a project on 12th to 18th C AD Lithuanian dogs (MNI 198), carried out in 2019–2022. During the project, we assessed the dogs’ size, type, health, nutrition, and other characteristics. The results of the biometric studies of canines, including the development of their size and type, are published in other works [14,15,16].

This paper focused on the nutrition levels and related health issues of the dogs, reporting the results of the carbon (δ^13^C) and nitrogen (δ^15^N) stable isotope analyses of 75 individuals, dating back to between the 12th and 18th C AD. On the basis of stable isotope data and osteometric analysis, we discuss the dietary patterns of different sizes and types of dogs from sites related to different social statuses and time periods in the territory of present-day Lithuania, shedding light on the following issues:

The analyzed dogs were sourced from a variety of places—cemeteries, large and small towns, and castles. The studied sites belonged to different cultural spaces, namely the Grand Duchy of Lithuania and the Teutonic Order. This made it possible to assess whether the nutrition of dogs found in diverse geographical, social, and cultural environments differed. The sampled canines represented two time periods, namely medieval and early modern, enabling the assessment and evaluation of canines’ diets across 600 years. A large number of individuals (MNI 34) from a single site, Vilnius Lower Castle, made it possible to test if the diet of elite dogs of different sizes differed. We also compared the results of the dogs’ stable isotope analysis with those of humans of different social strata, as well as with the stable isotope data of omnivorous pigs.

The study starts with a background data regarding all sites used, as well as the description of the sizes and types of dogs found there, their health condition, and historical records on the diet of the dogs. These are the key data needed to understand the hypotheses we have assessed in this study on dog nutrition.

## 2. Background Data

### 2.1. Site Description

Vilnius Lower Castle was the central residency of the Grand Duke in the capital of the Grand Duchy of Lithuania from the early 14th century to the middle of the 17th century AD. The zooarchaeological finds, dating back from the 13th century to the middle of the 14th century, reflected the stage of the construction of the castle, and from the late 14th century to the 15th century represent the period of its prosperity. In the early 16th century, on the site of the castle, a new palace for the Grand Dukes of Lithuania was built, and this complex survived until the late 17th century. The castle was abandoned after a Muscovian attack in middle of the 17th century AD, and was completely demolished in the beginning of the 19th century AD. Canines analyzed in this study were found during the 1988–2014 excavations, in the cultural layers, which dated to between the 13th and 17th century AD [25]. In the Vilnus Lower Castle, 590 dog bones and bone fragments (MNI 51) were found. In four cases, almost complete skeletons were recovered. Considering the castle’s developmental stages and the clear stratigraphy, dogs’ remains were divided into two chronological groups for further research as follows: 13th to 15th C AD and 16th to mid-17th C AD.

Kernavė is one of the most widely explored medieval archaeological sites in Lithuania. Archaeological research has been carried out here since 1979. In the Middle Ages, Kernavė had a common defensive system of four or five mounds, and the residency of the Grand Duke of Lithuania was located on the hillfort of Altar (Lith. Aukuras) Hill. The hillfort complex was surrounded by the Upper Town of Kernavė on the northern side, while the Lower Town was situated to the south, in the Pajauta valley. Both the Kernavė castle and town were attacked by the Teutonic Order and burned down in 1390; neither were ever rebuilt [26,27]. Dog bones collected in the 13th–14th C AD cultural layers of the Upper Town of Kernavė and the hillfort of Altar (Lith. Aukuras) Hill were included in this research.

Masteikiai is the only burial site included in this study. Masteikiai cemetery was excavated in 1993–1994, and 68 human inhumations and cremations, as well as 62 horses’ burials were found. Most of the graves were disturbed, and osteological material was poorly preserved. Some of the burials (about 25) contained the whole horse, while others only the head and lower parts of the limbs. Canines’ remains were found in at least four horse graves. The part of the dog skeleton found in horse grave 26 was dated back to 1163–1265 cal AD. Three horse graves contained dog mandibles. The mandibles of two more canines were collected from disturbed burials. According to the archaeological finds, the cemetery dates back to the 12th–14th C AD [28,29].

Klaipėda (formerly Memelburg) castle is located in western Lithuania. It was built in 1252–1253 by the Livonian Order, and, in 1328, was passed over to the Teutonic Order, which made it the Order’s northern-most castle in Prussia. In 1257–1258, Memelburg was granted Lübeck city rights; however, at the end of the 13th century, further urban and economic development failed due to wars with local Curonians, Samogitians, and Lithuanians [30]. The dog remains discussed in this paper were excavated in 2016 in the northern part of the castle [31,32], from the cultural layers of late 13th–early 14th C AD and late 13th–16th C AD. δ^13^C and δ^15^N analysis was performed for dogs dated back to the late 13th–early 14th C AD.

The research also included early modern period dogs from larger cities and small towns. The largest number, twelve individuals, came from five locations in Vilnius town. Five canines were examined from three places in Kaunas, and two more specimens were examined from present-day Klaipėda, formerly Memelburg town, belonging to the Duchy of Prussia. One animal was examined from the small coastal town of Palanga, situated close to Klaipėda; however, Palanga was part of the Grand Duchy of Lithuania. Three more dogs were analyzed from the small towns of Kernavė, Joniškis, and Anykščiai (Figure 1, Table 1 and Table 2).

### 2.2. Size and Type of the Dogs: Changes for 600 Years

Between the 12th and 18th C AD, dogs of different morphotypes were bred in the territory of present-day Lithuania, and their greatest variety was found in medieval castles. The highest diversity of canines was recorded in the medieval Vilnius Lover Castle. There were small-sized ‘toy’ type dogs, tall and slender sighthounds, large and robust molossians, and spitz-type individuals of all builds. The overall height of the castle dogs in the 13th–15th C AD varied from 28 cm to 75 cm (avg. 53.4 cm). In the later phase of the castle (16th–17th C AD), canines’ sizes ranged from 31 cm to 74 cm (avg. 50.2 cm).

In the early modern period, urban dogs in large towns were 53.1–54.3 cm on average. Meanwhile, the smallest animals were found in the early modern period Memelburg/Klaipėda town and in small towns of Lithuania, with an average height of 45.9 cm and 47.3 cm, respectively (Figure 2). A surprising number of quite large dogs were found in Lithuania; this is surprising as smaller canines were usually found in the surrounding areas during the discussed periods, especially in urban areas [3,7,14,33].

In the early modern period, the variety of dog types declined, with very small dogs no longer being found, and fewer and fewer large dogs, sighthounds and large Molossians, being available. Meanwhile, medieval sites of Kernavė and Vilnius Lower Castle contained a considerable number of sighthound type canines—gracile, tall, slender, long-headed, narrow-snouted individuals. However, they have only been found in medieval sites that are generally related to the highest social stratum. The sighthound-type canines were almost absent from the Klaipėda/Memelburg castle of the Teutonic Order, characterized by its large and robust dogs. Only a few large Molossian-type dogs have been found in the rest of Lithuania [14,15].

Returning to sighthound-type dogs, these were widespread in Lithuania since at least the 13th century. Sighthounds were often bred by the elite until around the 16th C AD. However, from the 16th C AD onwards, there is a clear decline in the popularity of this type of dog. Despite this, sighthounds maintained their exceptional symbolism of noble status until at least the late modern period. This is evident from the decrease in sighthound prices mentioned in historical records; sighthound-type dogs are also absent from zooarchaeological materials from the early modern period. Moreover, historical records from the 19th C AD also reveal that sighthounds were very rarely bred in Lithuania [14].

In early modern period, there is a remarkable change in dog morphotypes. Medium- and large medium-sized individuals with broader skulls and shorter snouts prevailed in the 16th–18th C AD. At that time, the largest dogs were urban canines, while smaller animals lived in the small towns and in Vilnius Lower Castle. Such differences in the size might be caused by the different uses and functions of the dogs [14,15]

The types of dogs recorded in the early modern period zooarchaeological material and the tendencies of their development correspond closely to the types of canines bred in Lithuania and those described in press and pictures from the 19th C AD, which perfectly reflect the preferences of dog selection at the time (Figure 3 and Figure 4).

The main criteria for selecting dogs in Lithuania and nearby countries in the 19th C AD were their working abilities. There is no doubt that the same patterns existed in earlier centuries. As historical data have revealed, previously popular large dogs, e.g., large sighthounds and robust Molossian-type hounds, were almost completely extinct in Lithuania by the end of the 19th C AD, with some Molossian-type canines having been extinct even earlier. The main reason for the loss of large dogs was the near extinction of large game in the 19th C AD. Instead, small game and birds were hunted, and thus large canines were simply no longer needed. In addition, in the 19th C AD, new breeds of hunting dogs, such as setters and spaniels, were being bred in western Europe, and had arrived in the Baltic region. These new dogs had excellent characteristics and were very suitable for hunting small game, thus becoming very popular in the Baltic region, as well as in Poland. These two reasons were perhaps the most important reasons for the extinction of a number of old native dog breeds in the 19th C AD, which had previously been bred in areas like present-day Lithuania, as well as Latvia and Poland. These included the large native Lithuanian hound ogar (Figure 3), the Curonian hound, and the Curonian pointer, as well as Curonian sighthounds and several other types of hunting dogs [14,15,34,35].

The sizes and types of dogs investigated, their development in the medieval and early modern period, as well as the diversity of canines in the sites of different periods and statuses are very important for understanding the diet patterns of dogs. Namely, the macroscopic analysis of the skeletal remains of dogs, together with the biometric data obtained and the observed differences in the health statuses of animals from different sites, has influenced our hypotheses regarding the dietary specificities of dogs of varying sizes, as well as of individuals living in different time periods and in different environments. These hypotheses were assessed using stable isotope analysis, and the results are presented in this study.

### 2.3. Diet of Elite Dogs: Historical Data

Information regarding the diet and care of elite dogs has been provided by inventory books and other historical records. The most detailed information about dog keeping, feeding, and training can be found in the fundamental work by Jan Ostroróg, The hunting with hounds (original title in Polish: “Myślistwo z ogary”). This work was published for the first time in Krakow in 1618 [23].

Most elite canines were highly expensive hunting dogs, who had to be in perfect physical form during hunts. Therefore, the nutrition and care of elite dogs was of great importance; they was certainly better than those of many urban animals [14]. However, this did not mean that the diet of elite hunting canines was rich in meat. Moreover, the rule that dogs should be hungry before a hunt is still common today. The diet of elite dogs was based on oat porridge, and was usually supplemented with fat, blood, and meat waste; however, the animals were also fed a variety of meats such as lamb, pork, and poultry. Bread was also sometimes bought or specially baked for dogs, and they were also provided with salt. Manors and castles had special kitchens for preparing dog food. During hunts, these animals were given the entrails and other waste products of game animals [17,23,24,36]. Jan Ostroróg wrote that hunting canines should only be given oats and no other cereals, especially barley, as barley reduces a dog’s sense of smell. However, to help the individual gain weight, a little rye should be added to the oat porridge. He also pointed out that goat meat is highly recommended for canines [23]. The same traditions and regulations for dog feeding were still present in the 19th C AD. This indicates that hunting with, breeding, and caring for dogs (and other animals) in Lithuania, recorded in detail in the 19th C AD [34,35], had not changed much since the 16th–17th C AD, or perhaps even earlier.

## 3. Material and Methods

### 3.1. δ^13^C and δ^15^N Analysis and AMS ^14^C Dating

The minimum number of individuals (MNI) analyzed during the project on Lithuanian dogs was 198. Of these, MNI 87 (43.9%) dated to the medieval period, with the majority of the canines being found in Vilnius Lower Castle (MNI 46). Fewer specimens were recovered from Kernavė town and hillfort (MNI 21), Klaipėda/Memelburg Castle (MNI 13), and other sites (Figure 1, Table 1). The number of dogs from the early modern period was slightly higher—MNI 111 (56.1%). This group mainly consisted of early modern animals from large and small towns, dating back to between the 16th and the 18th C AD (MNI 79), with fewer dogs from Vilnius Lower Castle (MNI 28) and other sites (Figure 1, Table 1 and Table 2).

In total, bone collagen carbon (δ^13^C) and nitrogen (δ^15^N) stable isotope analyses were conducted on 75 dogs. Samples of individuals from ten medieval and early modern sites were selected, including two castles, a medieval site, a hillfort, three large towns, and three small towns (Figure 1; Table 1). A total of 45 medieval and 30 early modern dog samples were analyzed. Of these, eight individuals were directly dated, and AMS dates are published for the first time in this study (Table 2). The largest dataset of stable isotope data came from Vilnius Lower Castle medieval canines (MNI 27) and early modern Vilnius urban dogs (MNI 12). In other sites, the number of analyzed individuals varied from one to nine (Table 1 and Table 2).

Carbon (δ^13^C) and nitrogen (δ^15^N) stable isotope analysis and AMS ^14^C radiocarbon dating were carried out at the Center for Physical Sciences and Technology in Vilnius, Lithuania (FTMC). Bone collagen extraction was performed according to the acid–alkaline–acid (AAA) procedure, followed by gelatinization [37]. The samples were treated with 0.5M hydrochloric acid and 0.1M sodium hydroxide. Bone collagen was gelatinized in a pH3 solution at 70 °C for 20 h. The gelatin solution was filtered through an Ezee filter and freeze-dried. The bone collagen carbon and nitrogen stable isotope ratios were measured using an elemental analyzer (Thermo Fisher Scientific, FlashEA 1112, The Netherlands) which was connected to an isotope ratio mass spectrometer (Thermo Fisher Scientific Finnigan Delta Advantage, Bremen, Germany). Repeated analyses of laboratory working standards provided standard deviations (SD) of less than ±0.1‰ for δ^13^C and ±0.15 ‰ for δ^15^N values. Stable isotope data are expressed as δ values in permil (‰), relative to the international standards V-PDB (Vienna Pee Dee Belemnite) for carbon isotopic values and atmospheric air for nitrogen isotopic values. To ensure that the isotopic signals were not affected by diagenetic processes and accurately reflect the dietary changes, collagen quality parameters were calculated, the main parameter being the C/N atomic ratio, which fell in the range of 2.8–3.6, as suggested by many authors [38,39,40,41]. Conventional ^14^C dates were calibrated with the OxCal 4.4.4 software using the IntCal20 curve [42]. All the ^14^C dates are reported within the 95.4% (2σ) confidence interval. To describe stable isotope values for all samples, the arithmetic mean ± the standard deviation (SD) was used. Statistical calculations were performed using the R-commander package [43].

### 3.2. A Bayesian Model for Diet Reconstruction (FRUITS)

The diet reconstruction analysis was performed using FRUITS, a Bayesian mixing model for diet reconstruction, version 2.1.1 Beta (https://sourceforge.net/projects/fruits/files/, accessed on 17 March 2024). Calculations were performed based on the recommendations of Fernandes and colleagues [44,45]. The mean isotopic values of possible food sources were retrieved from published data. The mean values were as follows: animal-based food (grouped cattle and pigs), δ^13^C −21.8 ± 0.6‰ and δ^15^N 7.8 ± 1.3‰ [21,22]; wild game, δ^13^C −22.9 ± 0.7‰ and δ^15^N 4.7 ± 1‰ [21,46]; freshwater fish, δ^13^C −25.3 ± 1.9‰ and δ^15^N 9.2 ± 2.1‰ [21]; and cereals, δ^13^C −25.0 ± 1‰ and δ^15^N 4.7 ± 0.9‰ [47]. The cereal δ^13^C value was adjusted according to diet to tissue offsets from Fernandes and colleagues [45]. Fecal values were calculated according to Kuhnle and colleagues [48] and Reid and colleagues [49], on the basis of the bone collagen values of commoners living between the 16th and 18th C AD in Vilnius. The fecal values were δ^13^C −22 ± 0.4‰ and δ^15^N 8.5 ± 0.8‰.

### 3.3. Biometrical Analysis

Withers height, type, age, and pathologies were determined for all dogs sampled via stable isotope analysis (Table 2 and Appendix A). Zooarchaeological studies were carried out at the Zooarchaeology Laboratory of Vilnius University. Bones were measured according to A. von den Driesch [50]. Age estimation was based on epiphyseal fusion, teeth eruption [51], and wear stage [52]. Withers heights of the dogs were calculated according to the coefficients of R. A. Harcourt [2]. When the individual had more than one bone used to calculate their height, the average height was taken. Based on withers height, dogs were classified as one of seven categories [53,54]: dwarf (<25 cm); very small (25–29 cm); small (30–39 cm); medium (40–49 cm); medium large (50–59 cm); large (60–69 cm); and very large (≥70 cm). When height estimates were not possible from long bones, measurements of the mandible and skull were used to estimate the dog’s size [55]. According to the mandibular length, dogs were divided to five size categories: small (<113 mm), medium-small (114–124 mm), medium (125–135 mm), medium-large (136–146 mm), or large (≥147 mm). Based off of skull length, five size categories of dogs were estimated: small (<155 mm), medium-small (156–170 mm), medium (171–185 mm), medium-large (186–200 mm), and large (>200 mm). According to the skull index (greatest zygomatic breadth/total length), dogs were divided into three categories: dolichocephalic, mesocephalic, or brachycephalic [54]. Measurements in the indices were made according to von den Driesch [50]. The pathologies were determined by dog orthopedist-traumatologist, Dr. Valdas Vaitkus, former associate professor of Lithuanian Veterinary Academy.

Canines’ remains from Vilnius Lower Castle are stored in the National Museum—Palace of the Grand Dukes of Lithuania, and dogs from all other sites are stored in the Zooarchaeological Repository of Vilnius University, Faculty of History.

## 4. Results and Discussion

### 4.1. Health and Age of the Dogs: Zooarchaeological Data

The difference in the health, care, and diet of the urban and elite dogs was most evident in their dental and overall oral health conditions. The urban canines often suffered from dental and jaw problems such as severe tooth wear, tooth loss, and periodontal disease (Figure 5). In small and large towns, a third of the dogs suffered from the aforementioned oral health problems, while in castles, the number of animals suffering from such problems was half as high—approximately 14% of the individuals [14,15]. The higher frequency of oral pathologies in the urban dogs is most likely due to their high malnutrition and regular consumption of hard food (bones) [56]. In contrast, the dental and overall oral health of the castle canines were very good. This is likely due to the presence of good, specially prepared food, and the general good care of the expensive elite dogs. Meanwhile, skull injuries affected both the city and castle dogs. A similar number of healed skull traumas, mainly impact fractures of the frontal bone and maxilla (Figure 5 and Figure 6), were found in the urban and castle canines in Vilnius, with 24% and 22% of individuals, respectively [14,15].

The skull injuries could have been caused by a variety of events, such as fighting with other dogs, hunting, or being beaten by humans [54,56,57,58]. Only medium-sized and larger dogs were among the castle and urban dogs that suffered head traumas. A similar pattern prevailed in the present day, with medium-to-large dogs predominating in modern dogs who had sustained head traumas due to canine fights. The most common injuries sustained by the modern canines in a fight are to the head and chest (approximately 50% of cases). Meanwhile, 70% of all injuries occurred in male dogs, and the most commonly involved were young dogs of approximately two years of age. Conflicts decreased with age, as injuries of this kind were very rare in dogs over six years of age [59]. Following these patterns, we can assume that a higher number of the injured archaeological canines were likely males, and that a significant number of head injuries might have occurred during conflicts when young.

Contemporarily, approximately 88% of dogs survive to the age of five and 62% reach 10 years of age [60]. The situation was certainly different in the Middle Ages and the early modern period, as the low mortality of dogs today is mainly due to good care. Indeed, today, only approximately 50% of stray dogs in urban areas reach the age of one [57,61]. Thus, canine mortality in urban areas should have been even higher a few hundred years ago. The situation for elite dogs is somewhat difficult to describe, but it seems plausible that their situation was much better. On the contrary, canines suffered serious injuries and death during hunts, and many animals died of diseases. Dogs had a high mortality rate from a variety of diseases in the 19th and even 20th C AD. Canines were often infected with rabies or glanders, or suffered from ear and intestinal diseases, and, quite often, all of the dogs of an individual keeper died [34,62,63,64]. As far back as the 1930s, rabies was rampant in Kaunas city, leading to the mass killing of dogs [65].

Considering the age of the analyzed dogs, it can be stated that, with the exception of a few puppies and juveniles under a year old, almost all of the nearly 200 studied individuals were already adult animals. These were dogs with permanent teeth, fused epiphyses, and vertebral plates already grown. However, a closer look at the age of the canines revealed certain patterns. According to tooth wear, the majority of the dogs buried in the Masteikiai cemetery were young. Five out of six dogs could have been around 3–4 years old, with only dog, ID 19, being somewhat older. The same patterns in age were also observed in Scandinavia [6]. Young adult canines up to the age of 4–6 years were also predominant at Vilnius Lower Castle and Kernavė. Meanwhile, many more dogs with badly worn teeth were found in the urban areas of Vilnius and Kaunas, as well as in small towns. As mentioned above, the heavy wear of the teeth may have been the result of poor nutrition. Meanwhile, 19th C AD descriptions of hunting often mentioned that dogs of one or another “breed” serve only for a short time, due to the very intense workload involved in hunting [34,35,66]. Indeed, maybe the castle canines were less likely to survive to older ages. However, this assumption was not supported by the scarce postcranial skeletal pathologies of the castle dogs.

### 4.2. Dog Nutrition between the 12th and 18th C AD: Stable Isotope Results

#### 4.2.1. General Data: Diet at Different Times and in Different Places

As might be expected from the study of 75 individuals from different periods and sites, the nutrition of the 12th–18th C AD dogs appeared to be non-homogeneous, with variations relating to both the period in which the dog lived and the social context of the site. The δ^15^N range of all dogs was from 7.5‰ to 13.1‰, with a mean of 10.3 ± 1.3‰ (Table 2 and Table 3). Keeping in mind the range of ~3.6‰ in one trophic level for nitrogen and 1–1.5‰ for carbon [67], these data exhibit a 1–1.5 trophic-level range, according to the individual.

The lowest δ^15^N values were measured for dogs from all of the medieval inland sites—Masteikiai (avg. 9.5‰), Kernavė (avg. 9.2‰), and 13th–15th C AD Vilnius Lower Castle (avg. 9.8‰) (Table 3, Figure 7, Figure 8 and Figure 9). The nitrogen stable isotope values of the canines found in these medieval sites were significantly different from those of the early modern period dogs (*p* < 0.05). In general, such nitrogen values are typical for omnivores, as plant food was the main contributor to their diet, with the occasional addition of meat [10,18,21,22]. The highest mean δ^15^N values were recorded in dog samples from Klaipėda/Memelburg Castle (avg. 11.6‰). Meanwhile, the average nitrogen value of the 16th–17th C AD animals from Vilnius Lower Castle (10.5‰) was almost 1‰ lower than that of the Vilnius urban dogs from the same period, but 0.8‰ higher than the castle canines from the 13th–15th C AD (Table 3, Figure 7). However, there was no significant difference between the δ^15^N values of the dogs found in these sites (*p* = 0.05, Welch two-sample *t*-test).

Among the inland sites, the average δ^15^N value of the Vilnius urban dogs was the highest (11.4 ± 0.8‰), which was 1.7–2.3‰ higher than for the dogs from medieval sites (*p* < 0.05) (Table 3, Figure 7 and Figure 9). Kernavė canines had the lowest nitrogen values, most likely due to the mostly vegetarian diet (9.2 ± 0.9‰). Relatively higher δ^15^N values in animals and humans may be due to several factors, such as high meat or fish consumption, or even the opposite—prolonged starvation. The high δ^15^N values indicate that the plants were fertilized with manure, as those animals who were fed on these plants would also have had a higher nitrogen stable isotope ratio [68,69,70]. It is likely that the differences recorded and the lower δ^15^N values of the medieval dogs indicate that the diet of the canines from the earlier period contained a higher proportion of plant-based foods than in later times. On the contrary, the meat of game animals may have also contributed to the lower nitrogen values. Wild herbivores and omnivores have lower nitrogen values than domestic animals (Table 3, Figure 10). However, Bayesian modelling demonstrates very similar nutrition levels of the dogs from medieval sites and early modern Vilnius Lower Castle (see below). Elite hunting dogs were fed game meat during hunts. Given that hunting was frequent, it can be assumed that wild game meat may have contributed to the lower δ^15^N values of the medieval dogs. Meanwhile, the slightly higher nitrogen stable isotope values of the castle canines in the early modern period may be related to the declining popularity and importance of hunting in this period [14]. Thus, the nutrition of later dogs may have simply increased with the increased consumption of meat.

The relatively high δ^15^N values of the urban canines from Vilnius and Kaunas were likely not due to higher meat or fish consumption, but rather to the high proportion of feces and animal waste (i.e., bones) of city dwellers comprising their diet, as well as chronic starvation. Presumably, a high number of the urban canines were stray dogs, suffering from food shortages, and feeding on feces and bones [12,74]. Thus, their nitrogen stable isotope values would have been higher than those of elite dogs, who were fed on plant-based foods like porridge or game animals. The former, mostly hunting dogs, were well cared for and fed [14]. Meanwhile, free-ranging, scavenging urban dogs and pigs have been described eating even the bodies in cemeteries in a variety of towns in the early modern period, as well as in the early 20th C AD [33,65,75]. It is worth mentioning that, overall, the lowest and highest δ^15^N values were found in the Vilnius Lower Castle (13th–15th C AD) dogs. However, the highest value came from the aforementioned individual, ID 8, who was of exceptional size and build, and whose diet and life were, perhaps, remarkably different from those of the typical urban and even castle canines.

The dogs’ overall carbon stable isotopic values ranged from −21.9‰ to −19.9‰, and the average value was −20.7 ± 0.4‰ (Table 2 and Table 3, Figure 8 and Figure 9). The highest δ^13^C values were measured in the Masteikiai canines’ bone collagen samples (avg. −20.2 ± 0.3‰), while in Klaipėda/Memelburg Castle, these were the lowest (avg. −21.2 ± 0.7‰). However, the statistical analysis of the carbon stable isotope ratios did not show significant differences between sites (Kruskal–Wallis rank sum test, *p* > 0.05). The slightly higher carbon values of the Masteikiai dogs may reflect the presence of millet in their diet. In medieval Lithuania, millet was still cultivated in some areas, although only to a small extent [76].

#### 4.2.2. Diet of Differently Sized Dogs

One hypotheses is that there might be dietary diversity between dogs of different sizes. The assumption is that larger dogs might be fed “better”, and their diet may have contained more meat, which should be expressed via higher δ^15^N values. Urban dogs were not suitable for testing this hypothesis, as the environments of these canine varied and remain unclear. In addition, urban dogs were quite similar in size. Therefore, the most suitable animals for assessing the diet of differently sized canines were the pets from Vilnius Lower Castle.

It appeared that, with the exception of individual ID 8, generally, the diet of all of the castle dogs was similar. The smallest individuals, ID 5 and ID 12, had nitrogen stable isotope values of 10.4‰ and 10.9‰, respectively, which were slightly higher than the averages for the castle canines of the period (9.8‰) (Table 2 and Table 3, Figure 10), but these differences were not statistically significant (Kruskal–Wallis chi-squared test, *p* > 0.05).

As mentioned above, the only dog with a significantly different nutrition (*p* < 0.05) was individual ID 8 (1302–1410 cal AD). This animal was 74.3 cm tall, with a gracile build and a nearly ultradolichocephalic cranium shape. The skull of this individual, according to the cranium and palatal indices (length/width ratio), was almost identical to the modern Russian wolfhound, and was highly similar to the English greyhound. Its oral health and the condition of its teeth were very good; however, the right humerus showed signs of tendinitis, the shoulder joint was damaged (osteochondritis dissecans), and the fibula of one leg was fused to the tibia due to diaphysis [14]. The skeleton of this dog was found almost intact; it is likely that the individual was carefully buried. Its δ^13^C value was −20.1‰, and its δ^15^N value was 13.1‰, i.e., its nitrogen isotope value was 3.3‰ higher than the average of the castle dogs of the same period. This might indicate that ID 8 consumed more meat or fish than the other canines. It is hardly likely that the high nitrogen value of this dog was due to long-term starvation, or that the dog consumed a lot of fish. Fish is not mentioned in the diet of the elite canines of the Grand Duchy of Lithuania, nor in Ostroróg’s work [23] or the historical sources of the 19th C AD.

Fish was not important in the diet of the pagan Lithuanian population [71,77,78]. It is therefore more likely that dog ID 8 consumed a lot of meat, most probably from domestic animals. Its exceptional diet would therefore confirm the assumption that this sighthound was an exclusive pet with a special status. The high status of sighthound-type dogs since prehistoric times has been confirmed by a number of studies [14,15].

Sighthounds were still elite dogs in Poland and Lithuania, and were bred more commonly in the first half of the 19th C AD. However, in the second half of the century, they were already very rarely bred [34,35]. As zoologist and expert on hunting dogs and the hunting of the Russian Empire, Leonid Sabaneev, wrote in his magazine, Hunter’s Calendar, purebred Polish sighthounds were almost extinct in Poland at the end of the 19th C AD and could only be found in the territory of modern-day Ukraine. One of the reasons for this decline in sighthounds in Poland is the tax of 15 rubbles, which was introduced for the keeping of a sighthound after the Polish–Lithuanian uprising against the Russian Empire in 1863 [35] (p. 101). The decision of the Empires’ regime to apply tax to the breeders of sighthounds—nobles—once again confirms the great importance of this type of dog as a status symbol and the long-lasting tradition of this symbolism.

Meanwhile, the stable isotope composition of the other largest dogs, sighthound-type individuals ID 1 and ID 2, or the large Molossian-type canines ID 9 and ID 11 did not differ (*p* > 0.05) from that of the other dogs (Figure 10, Table 2). In addition, the entire skeleton of dog ID 11 was also recovered, and the dog was very large (74.3 cm) and extremely robust. Thus, the results would suggest that the dog’s exceptional size or morphotype did not necessarily lead to its superior diet. On the contrary, it is difficult to say to what degree the size or morphotype may have contributed to the dog’s status, special ranking, or to the resulting exceptional diet, as compared to other dogs. For example, one of Ostroróg’s recommendations was not to feed the best hunting dog first, even if they wanted to, because a good dog should not be fat [23]. Most likely, the exceptional diet came from not living with other dogs in the kennel, but, for example, living close to its owner in the position of a companion. Such dogs certainly existed [17], and thus we can assume that dog ID 8 was one of them.

#### 4.2.3. Diet of Coastal Dogs

The diet of the dogs at Klaipėda/Memelburg Castle should be discussed separately. As mentioned above, the canines found here displayed the highest nitrogen and lowest carbon stable isotope values. The δ^13^C values of the Klaipėda/Memelburg canines were significantly different from the dogs sampled from the other sites (*p* < 0.05) (Table 2 and Table 3, Figure 7, Figure 8, Figure 9 and Figure 10). The exceptional data regarding the animals from the Teutonic Order castle were likely due to completely different reasons than for those individuals found in urban areas. The high nitrogen and low carbon values probably reflect the presence of freshwater fish in the diet of the dogs from this coastal castle. FRUITS modelling also suggest a higher intake of freshwater fish in coastal dogs compared to the other sites (Figure 11 and Figure 12). Klaipėda/Memelburg Castle is located near the sea and the Curonian Lagoon, meaning that fishing, mainly in the lagoon, was an important part of the economy of the castle and the inhabitants of the town [79]. A large number of fish remains have been found in Klaipėda/Memelburg Castle, of which freshwater fish predominate (99.5%), with the most abundant species (NISP 37.8%) being zander (*Sander lucioperca*) [80]. Fish and their waste could be consumed by castle dogs in huge amounts. This practice was also evidenced by canines from coastal sites of other periods, where freshwater fish made up a very significant part of their diet. For example, the δ^13^C of one dog from the Subneolithic Šventoji 43 site was −23.8‰, and the δ^15^N was as high as 13.9‰ [19] (p. 91–93).

However, the diet of the Klaipėda/Memelburg Castle dogs was not homogenous. This is reflected by the wide range of carbon and nitrogen stable isotope values [Table 2 and Table 3, Figure 7, Figure 8, Figure 9 and Figure 10], and may be related to the different functions or origin of the dogs analyzed. The nitrogen value (9.9‰) of one individual (ID 21) was approximately 2‰ lower than that of the other dogs found in Klaipėda Castle, who did not differ from those found in the medieval inland sites of Masteikiai, Kernavė, and Vilnius Lower Castle. The carbon isotope value (−21.9‰) of ID 21 was lower than those of the inland canines. The carbon and nitrogen stable isotope values compared to the other Klaipėda dogs would suggest a lower presence of fish and meat in the diet of ID 21. Another dog, ID 22 (δ^13^C −21.8; δ^15^N 11.0), found in Klaipėda/Memelburg Castle, was also similar in the diet of ID 21 (Table 2, Figure 10). Differences in the nutrition of the Klaipėda/Memelburg Castle dogs would suggest that animals ID 21 and ID 22 may have been non-local, potentially having formerly lived on the inland. That these inland dogs did not feed on fish is suggested by both their stable isotope values and their historical data. In addition, research has also indicated a nonsignificant contribution of fish to the diet of medieval and early modern populations in Lithuania [71,72,73]. However, mixing modelling did not demonstrate any significant differences in the food sources of individuals ID21 and ID22 when compared to the other canines (Figure 12).

Canines with an inland diet could reach the castle with their owner or as a gift. Dogs were a popular gift in both the Middle Ages and the early modern period. A number of medieval and early modern written sources provide evidence, confirming how dogs have been gifted over long distances [17]. Records from the mid-19th century suggest that, at the time, a hunting canine often changed its owner three to four times during its lifetime. Most often, the dog was given as a gift and later re-gifted, with hundreds or even thousands of kilometers separating the owners [34]. On the contrary, the exceptional diet of a few of the individual dogs may have been related to the specific status and function of the animals, e.g., they might have been hunting dogs, with a high intake of game meat, which resulted in relatively low nitrogen and carbon isotopic values. Moreover, dog ID 21 was of a somewhat exceptional size, being one of the most robust of all of our investigated canines from the 12th–18th C AD (Appendix A).

#### 4.2.4. Nutrition: Dogs vs. Pigs and Humans

The diet of urban dogs and pigs is often compared, as these animals shared similar habits and lifestyles. They wandered freely in towns, scavenging for their food, consuming a variety of waste and feces [33,75]. However, the stable isotope analysis of the dogs and pigs revealed that most of the pigs had a more plant-based diet when compared to the dogs, as their average carbon and nitrogen stable isotope values were −21.7‰ and 7.9‰ (Figure 10, Table 3). Similar tendencies have been observed in medieval and early modern Estonia [7]. Meanwhile, when compared to humans, the diet of the early modern canines in general was similar to that of the rural population during the same period (Figure 10, Table 3). However, the medieval dogs had slightly lower nitrogen and carbon stable isotope values than the rural human population, possibly due to a more plant-based diet and/or the consumption of game meat. Elite and urban dwellers, however, had higher nitrogen stable isotope values and lower carbon stable isotope ratios when compared to the dogs. These differences could be explained by the higher consumption levels of freshwater fish in humans. Fish was important in the diet of elites, but it did not play a big role in the diet of ordinary urban dwellers [71,73,81].

### 4.3. Bayesian Modelling

According to the FRUITS mixing model, cereals, a possible food source of dogs, vary from 11 ± 7% (urban Vilnius and Kaunas) to 47 ± 12% (Masteikiai). This food source makes the highest contribution in five out of seven dog groups from selected sites. In general, cereals’ contribution was similar in all sites, except in early modern urban contexts (Figure 11 and Appendix A). Estimated domestic ungulate (cattle and pig) contributions vary from 22 ± 8% in Klaipėda/Memelburg castle to 40 ± 16% (urban, Vilnius); however, in all medieval inland sites and early modern period Vilnius Lower Castle, the average distribution was almost the same, being 26 ± 6%. The estimated terrestrial game animal contribution was also similar across all sites, varying from 13 to 14 ± 6%. However, wild game was not included in modeling the urban dogs’ diet, as wild game in general is absent from urban zooarchaeological material. The hunting was the privilege of elites in the early modern period; therefore, the town dwellers generally did not hunt [14,24,25]. Freshwater fish intake was also similar in all inland non-urban sites, ranging around 5–6 ± 4%. It was slightly higher for urban dogs (8–9 ± 6–7%). Freshwater fish was more important in coastal site Klaipėda, consisting of 19 ± 8%. The lowest fecal consumption was in Klaipėda/Memelburg Castle (7 ± 5%), and was slightly higher in inland castles and other medieval sites. Meanwhile, in both early modern period urban contexts of Vilnius and Kaunas, feces consumption was almost the same (40 ± 17%) (Figure 11 and Appendix A).

Mixing modelling of the diets of dogs in Klaipėda/Memelburg would demonstrate the slightly diverse dietary patterns of some individuals, e.g., ID20. Its consumption is characterized by a small range of SD values, the highest mean intake of cereals, a slightly higher amount of domestic animals, and a low intake of wild game. However, no clear differences could be observed between the dietary patterns of Klaipėda dogs. ID21, with the lowest nitrogen and carbon stable isotope values, did not have an exceptional intake when compared to the other individuals; however, ID61 did, with the highest δ^13^C values (Figure 12 and Appendix A).

The mixing modelling demonstrates that the consumption patterns of dogs in medieval sites and early modern period Vilnius Lower Castle were very similar, with about 45–50% of the protein coming from cereals, 25% from domestic ungulates, 13% from wild game, and up to 10% from fish and feces. Fish consumption was the highest in the coastal Klaipėda/Memelburg Castle. In the diet of urban dogs, feces and domestic animal (bone waste?) might have accounted for 40%. However, it can be noted that the range of values is very large. In addition, different food sources and priors were selected for the different groups of dogs, based on the historical data of the dogs’ diets, teeth conditions, and oral health of the canines analyzed, as well as the zooarchaeological data, and the archaeological and historical context of the sites. These selections had clear impacts on the modelling results.

Other factors also played a significant role in the modelling results. Estimating fecal consumption is challenging, as fecal stable isotope values depend on a variety of factors, such as human diet (vegetarian, fish or meat based), exposure time in the open environment, etc. [12,48,49,74]. As there are no stable isotope studies of medieval and early modern coastal populations in Lithuania, the fecal values of Vilnius urban populations were included in the modelling of the Klaipėda/Memelburg dog intakes. Furthermore, dogs eat not only human feces, but also feces from other animals (horses, cattle) that were widely available for free-ranging dogs. However, assessing the contribution of different types of feces to the diet of dogs is hardly possible. We have not included animal fecal values in our modelling. In addition, it is likely that some dogs in Klaipėda/Memelburg consumed marine food (e.g., ID61), which we did not include in the model.

## 5. Conclusions

The reconstruction of dog diets and the analysis of stable isotope data presented challenges of interpretation. The main problem was the equivalence of the isotopic values, which can be caused by different factors. We used dietary mixing modelling as one of the tools to reconstruct dietary patterns. However, modelling the diets of omnivores that lived in very different environments and under very different conditions, which are only predictable to the researcher, can leave many possible scenarios and many unknown options. The available food source data, the archaeological and zooarchaeological contexts, and the historical data, as well as priors selected by researchers, have a strong impact on the modelling results. Modelling should therefore be treated with caution, and, in the case of the material analyzed in this study, the other sources of a different nature—biometry, health status, historical records—played a key role in both hypothesis development and testing, as well as in supporting the results of δ^13^C and δ^15^N stable isotope analysis.

The size, type, and health of canines from different time periods and sociocultural environments varied. This reflects the different living conditions, care, and nutrition of the dogs, as well as the diversity of their roles and functions in the daily lives of humans. Differences in the diet of medieval and early modern canines were caused by different factors relating to the social and cultural environments of the dogs. The stable isotope analysis supported the historical evidence, indicating that cereals were highly important in the diet of elite dogs. According to the historical records, oat porridge was the base of the diets of noble canines, and meat intake was low. In the Middle Ages, the consumption of plant-based foods might be higher compared to the early modern period. In the post-medieval period, changes in the diets of canines were observed, evident through increased nitrogen stable isotope values. For the Vilnius Lower Castle dogs, the shift could have been caused by an upturn in the intake of domestic animal meat and offal products in canine diets. In the case of urban dogs, the consumption of feces and bones, as well as starvation, might have been an important contributor to both their diet and health issues.

Most of the analyzed canines from the coastal Klapėda/Memelburg Castle of the Teutonic Order had a higher intake of fish in their diet. The patterns of their diet could be related to the location of the castle and the overall widespread consumption of fish in coastal areas. Furthermore, as the castle belonged to the Teutonic Order, the traditions involved in the diet of dogs might have differed as well. Nevertheless, the diet of the Klaipėda/Memelburg Castle canines was not homogeneous, and more inland diets of a few individuals may indicate a non-local origin or be related to a different function of the animals, leading to different feeding practices.

Dog size did not have a significant relation to the diet of the Vilnius Lower Castle canines. This may reflect the similar functions of the differently sized dogs, suggesting that the smallest dogs were not exclusive pets, but rather hunting canines. It could also suggest that, despite the different functions, animals were fed in the same way. In general, the nutrition of the studied dogs was similar to that of the rural human population from the late medieval and early modern periods. Meanwhile, when compared to pigs of the same period, the dogs tended to have a higher intake of animal meat in their diet.

## Figures and Tables

**Figure 1 animals-14-01023-f001:**
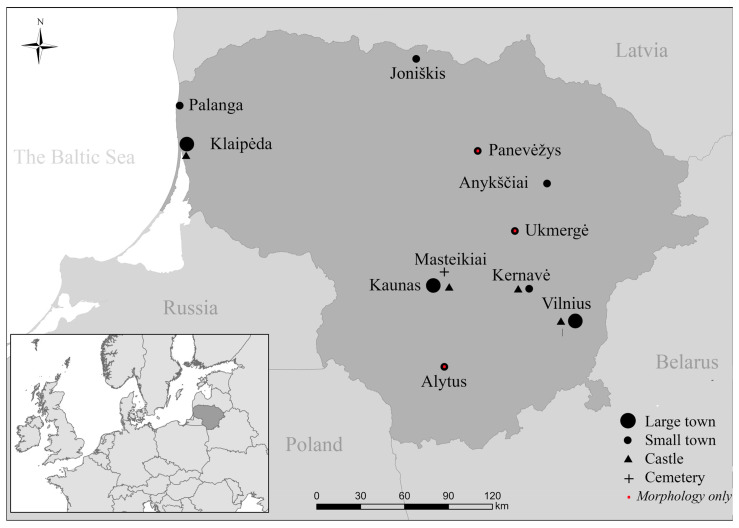
Location of the sites where dog remains were analyzed.

**Figure 2 animals-14-01023-f002:**
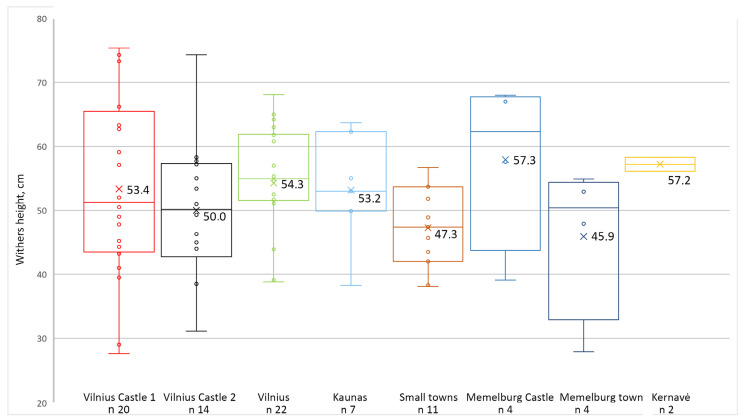
Withers height of dogs in the sites studied. Vilnius Castle 1—Vilnius Lower Castle 13th–15th C AD; Vilnius Castle 2—Vilnius Lower Castle 16th–17th C AD; Vilnius and Kaunas towns—16th–18th C AD; Memelburg Castle—recent Klaipėda Castle late 13th–16th C AD; Memelburg town—recent Klaipėda town 16th–18th C AD; Kernavė medieval town and hillfort 13th–14th C AD. For sites, their location, and more detailed biometrical data of the dogs, see Figure 1, Table 2 and Appendix A. Numbers indicate the average height of the animal; the line indicates the median [14].

**Figure 3 animals-14-01023-f003:**
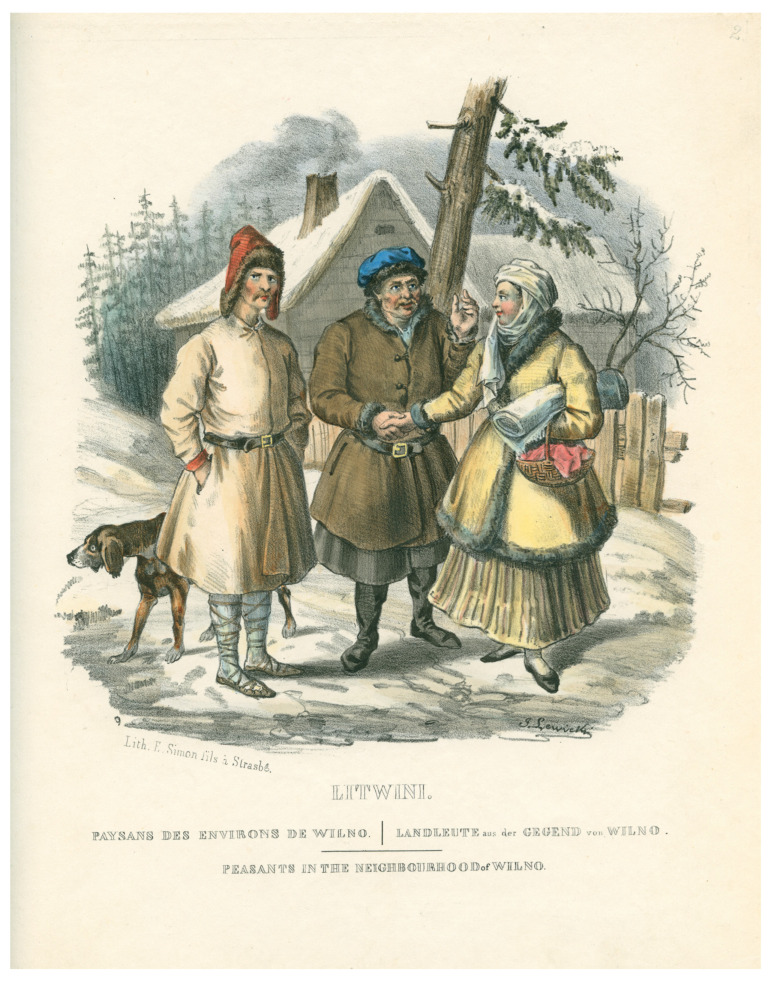
A dog similar to a large local hound (ogar), described in the literature and historical records of the 19th C AD, together with Lithuanian peasants in the Vilnius surroundings, 1841, LNM, IMik 4367.

**Figure 4 animals-14-01023-f004:**
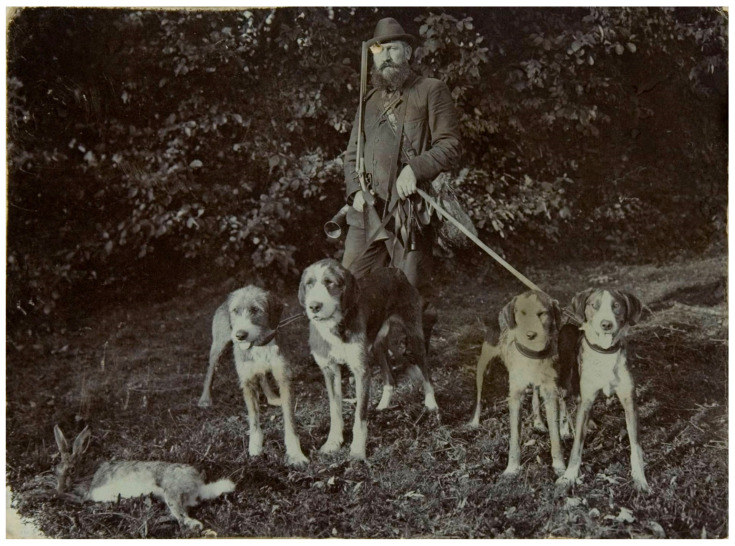
A hunter, Faustinas Olejarčikas, with dogs (1895–1904). Photo by Stanislovas Kazimieras Kosakovskis, 1895–1904. ©LNMMB.

**Figure 5 animals-14-01023-f005:**
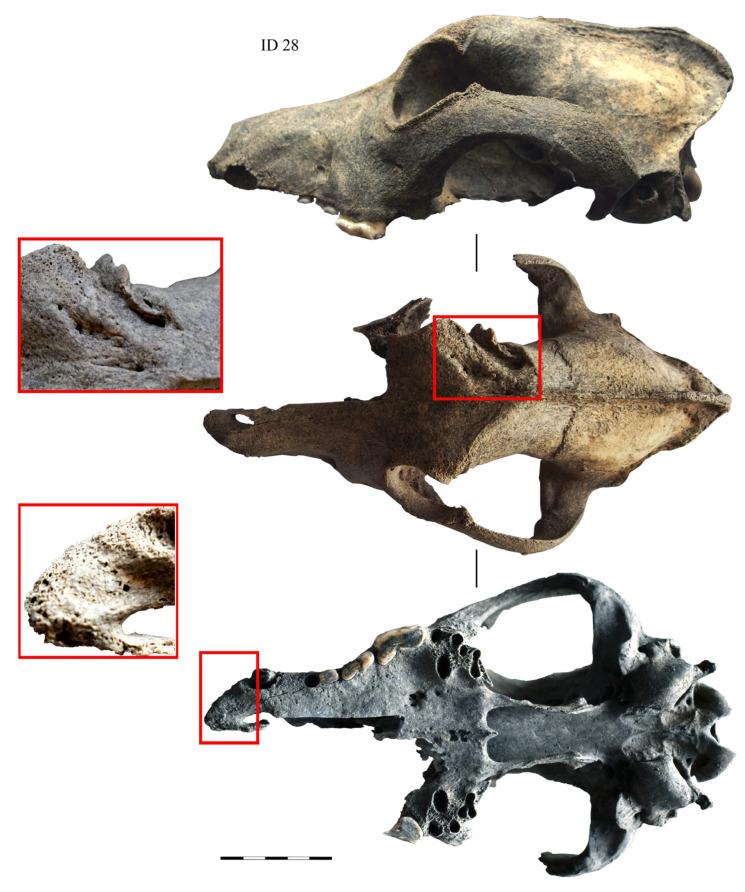
Dogs’ skull ID 28, found in Vilnius, Reformatai Square (16th–18th C AD). The animal suffered from alveolar inflammation, had lost some teeth, and had healed skull fractures.

**Figure 6 animals-14-01023-f006:**
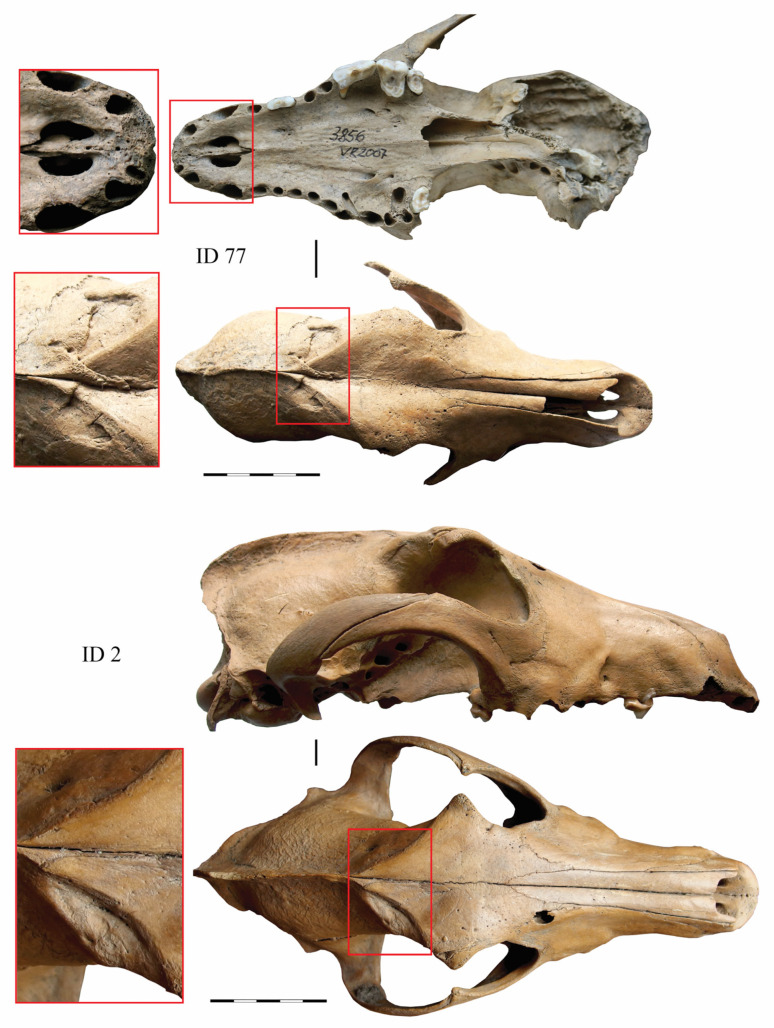
Skulls of two sighthound-type dogs, ID 2 and ID 77, with healed fractures from Vilnius Lower Castle, 13th–15th C AD.

**Figure 7 animals-14-01023-f007:**
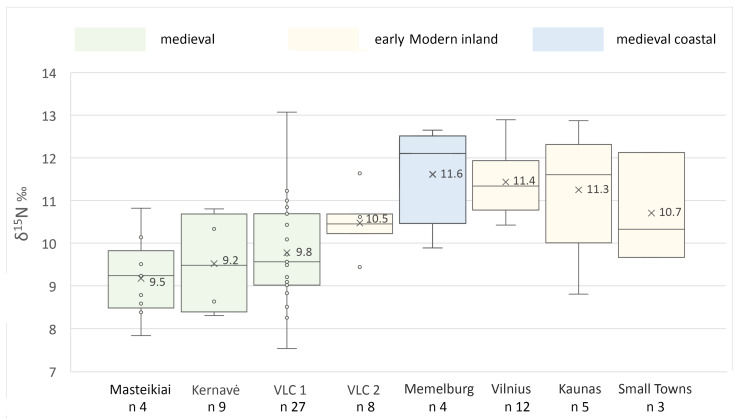
Distribution of the δ^15^N values of dogs from different sites and periods. Site explanations: VLC 1—Vilnius Lower Castle 13th–15th C AD; VLC 2—Vilnius Lower Castle 16th–17th C AD; Memelburg Castle—recent Klaipėda, late 13th–early 14th C AD; Kernavė—Kernavė medieval town and Aukuras hill hillfort, 13th–14th C AD. Numbers indicate the average value of stable isotope ratios; the line indicates median. For sites and their location, see Figure 1.

**Figure 8 animals-14-01023-f008:**
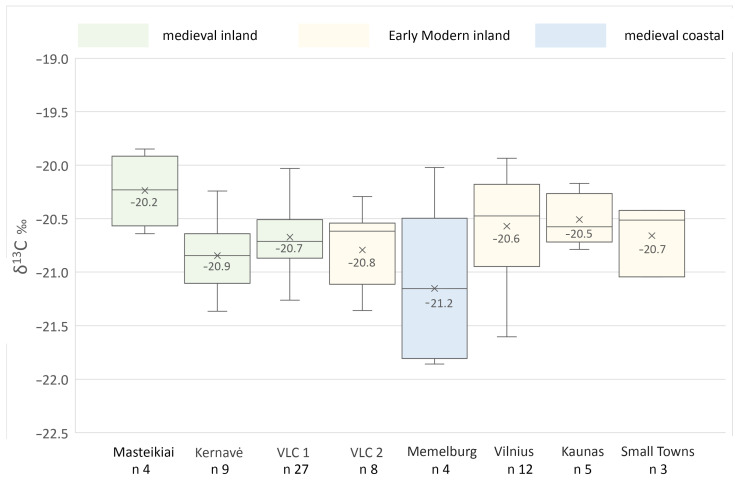
Distribution of the δ^13^C values of dogs from different sites and periods. For site explanations, see Figure 7. Numbers indicate the average value of stable isotope ratios; the line indicates median. For sites and their locations, see Figure 1.

**Figure 9 animals-14-01023-f009:**
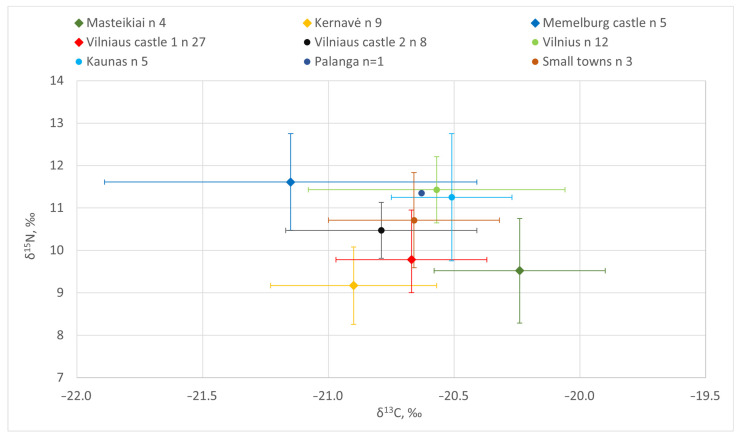
Carbon and nitrogen stable isotope ratios of dogs’ bone collagen samples from different sites and periods. Site explanations: Vilnius Castle 1—Vilnius Lower Castle 13th–15th C AD; Vilnius Castle 2—Vilnius Lower Castle 16th–17th C AD. Rhombs represent the medieval period; circles—the early modern period.

**Figure 10 animals-14-01023-f010:**
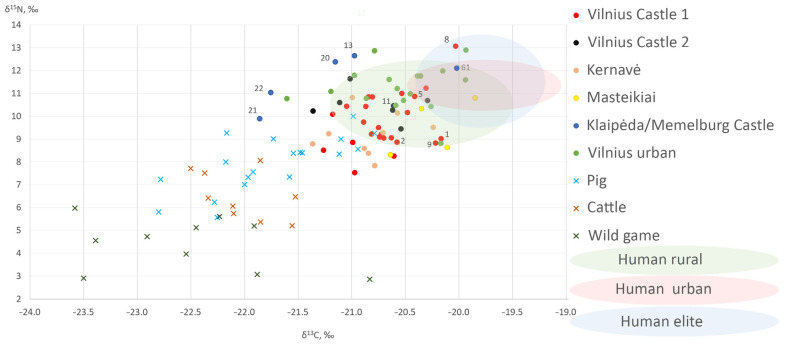
δ^13^C and δ^15^N values of dogs and other animals. Vilnius Castle 1—Vilnius Lower Castle 13th–14th C AD, Vilnius Castle 2—Vilnius Lower Castle 16th–17th C AD. Data for pigs, cattle, and wild ungulates from [21,22,46,71]; for humans, [21,71,72,73]. For more detailed data on references, see Table 3.

**Figure 11 animals-14-01023-f011:**
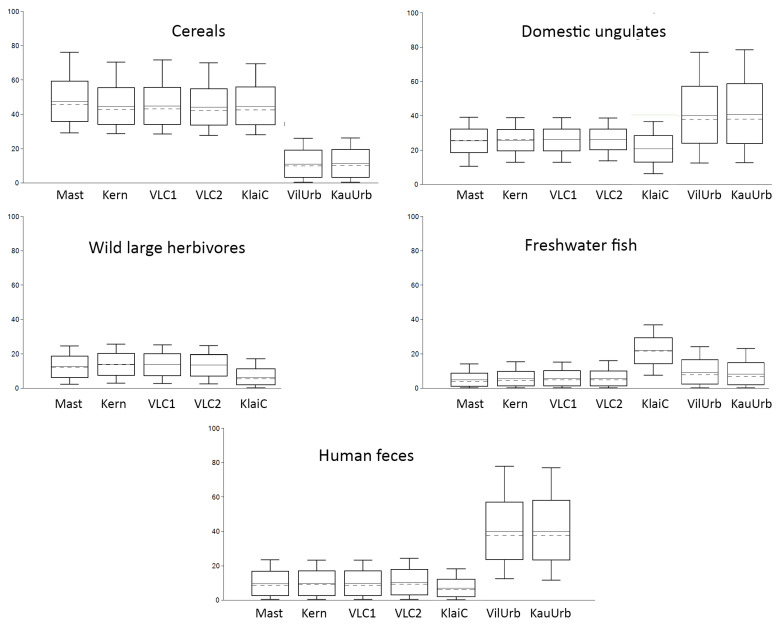
FRUITS mixing modelling of the possible food sources of dogs from the medieval and early modern sites. Priors for Masteikiai, Kernavė, Vilnius Lower Castle 13th–15th C AD (VLC1), and Vilnius Lower Castle 16th–17th C AD (VLC2): cereals > meat, cereals > game, cereals > fish, cereals > feces, meat > game, game > fish, meat > feces. Priors for Klaipėda/Memelburg Castle: cereals > meat, cereals > game, cereals > fish, cereals > feces, meat > game, meat > feces. Priors for urban (Vilnius (VilUrb) and Kaunas (KauUrb)): feces > fish, feces > cerels, meat > fish, meat > cereals.

**Figure 12 animals-14-01023-f012:**
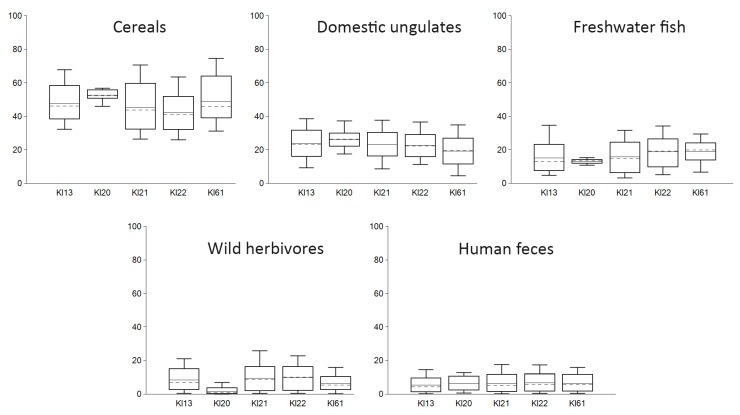
FRUITS mixing modelling of the possible food sources of dogs from Klaipėda/Memelburg Castle. Priors: cereals > meat, cereals > game, cereals > fish, cereals > feces, meat > game, meat > feces.

**Table 1 animals-14-01023-t001:** Number of dogs (MNI) analyzed in different Lithuanian sites, and the number of them sampled for stable isotope analysis.

Dogs/Sites	Medieval, 12th–15th C AD						Early Modern Period, 16th–18th C AD						
	Masteikiai Cemetery	Kernavė, Town and Hillfort	Vilnius Lower Castle	Klaipėda/ Memelburg Castle	Kaunas Castle	In Total	Vilnius Lower Castle	Vilnius, Urban	Kaunas, Urban	Klaipėda/ Memelburg, Urban	Small Towns	Manors	In Total
**MNI, morphology**	6	21	46	13	1	**87**	28	38	13	13	15	4	**111**
**MNI, δ^13^C, δ^15^N analysis**	4	9	27	5	0	**45**	7	12	5	2	4	0	**30**

**Table 2 animals-14-01023-t002:** Carbon and nitrogen stable isotope results and supportive information for dogs analyzed. List of dogs analyzed with biometric data of individuals. Skull category: D—dolichocephalic; SB—subbrachycephalic; SD—subdolichocephalic; SM—submesocephalic; UM—ultramesocephalic; Size SK—dog size category by skull; Size MD—dog size category by mandible; S—small; MS—medium-small; M—medium; ML– medium-large; L—large. Biometric data according to [17]. For bone measurements, see Appendix A.

No.	Dog ID	Site	Site Characteristic	Chronology, AD	Wither Height	Weight, kg	Skull Category	Size SK	Size MD	δ^13^C, ‰	δ^15^N, ‰	N%	C%	C/N	AMS ^14^C cal AD, 95.4%, 2σ	AMS ^14^C uncal BP	Lab. Code
1	72	Anykščiai, A. Baranausko sq.	Small town	16th–18th c.	38.3	14.0	DB	MS	M	−21.04	9.67	16.0	42.5	3.1			
2	69	Joniškis, S. Dariaus ir S. Girėno st.					UM	L		−20.42	12.12	14.6	38.2	3.1			
3	43	Kernavė, Vilniaus st. 24							L	−20.51	10.33	15.6	41.4	3.1			
4	57	Kaunas, Gertrūdos st. 51	Large town	16th–18th c.						−20.58	11.21	15.6	41.2	3.1			
5	75				55.0					−20.17	8.81	16.1	42.7	3.1			
6	71				62.3	27.0			L	−20.79	12.87	13.7	36.2	3.1			
7	58	Kaunas, Muziejaus st. 11			50.0	17.0				−20.36	11.76	16.0	44.4	3.2			
8	55	Kaunas, Perkūnas house sq.						S	S	−20.65	11.61	15.2	40.1	3.1			
9	7	Kernavė, Aukuras hillfort	Kernavė medieval town with the duke’s recidency on Aukuras hillfort	13th–14th c.				ML	L	−20.88	8.59	14.4	37.7	3.0			
10	45	Kernavė, Upper town			56.1	17.6				−20.71	9.28	15.9	42.7	3.1			
11	46				58.3	22.0				−20.78	7.84	16.1	43.3	3.1			
12	60								L	−21.37	8.78	14.0	40.7	3.4			
13	73							L	L	−21.22	9.24	15.6	41.7	3.1			
14	6							ML	L	−20.24	9.51	22.0	58.5	3.1			
15	44								S	−20.57	10.14	16.3	44.0	3.1			
16	23								M	−20.85	8.38	15.2	40.1	3.1			
17	24								M	−21.00	10.82	14.7	39.6	3.1			
18	22	Klaipėda (Memelburg) castle	German’s Order castle	late 13th–early 14th c.	68.0	30.0				−21.76	11.04	16.3	45.5	3.3			
19	61									−20.02	12.10	15.1	42.3	3.3			
20	13							L		−20.97	12.65	14.7	38.6	3.1			
21	20								L	−21.15	12.38	15.6	44.0	3.3			
22	21								L	−21.86	9.89	15.5	43.7	3.3			
23	32	Klaipėda (Memelburg), Kurpių st. 3	Town of the Duchy of Prussia	16th–17th c.	47.9	12.6				−20.40	9.90	12.7	35.9	3.3			
24	35				52.9	14.0				−20.50	9.50	12.4	35.3	3.3			
25	17	Masteikiai, disturbed grave	Cemetery	12th–14th c.						−20.11	8.63	15.9	44.3	3.3			
26	18	Masteikiai, disturbed grave							ML	−20.35	10.33	14.9	41.6	3.3			
27	19	Masteikiai, grave 22								−19.85	10.80	15.3	43.0	3.3			
28	16	Masteikiai, grave 26							L	−20.64	8.31	15.9	44.0	3.2	1163–1265	842 ± 28	FTMC-SN30-1
29	67	Palanga, J. Simpsono st.	Smal coastal town	16th–18th c.	53.9	20.6			ML	−20.63	11.35	15.8	44.2	3.3			
30	48	Vilnius Lower castle	Central castle of the Grand Duke	13th–15th c.	62.7	27.7				−20.58	8.52	16.0	42.0	3.1			
31	53									−20.31	11.22	16.0	41.9	3.1			
32	54									−20.70	9.04	16.5	42.7	3.0			
33	76									−20.84	10.85	15.7	41.7	3.1			
34	79									−20.74	9.09	16.2	42.2	3.0			
35	36				66.2					−20.59	10.43	16.4	43.5	3.1			
36	42				41.0					−20.99	8.85	18.3	48.5	3.1			
37	77							L		−20.61	8.26	19.1	45.6	2.8			
38	9							L		−20.22	8.83	15.2	39.6	3.0	725–660	750 ± 28	FTMC-PA75-4
39	5							S		−20.41	10.86	15.4	40.7	3.1			
40	51							MS		−20.72	9.21	15.7	41.7	3.1			
41	31				57.1	19.0			L	−20.78	9.56	15.3	40.3	3.1			
42	34								L	−20.87	9.49	17.5	46.1	3.1			
43	41								L	−20.81	10.85	16.7	43.2	3.0			
44	3				59.1	23.0	D	L		−21.26	8.51	15.6	39.8	3.0			
45	8				74.3	40.1	D	L	L	−20.03	13.07	14.5	38.3	3.1	649–540	592 ± 29	FTMC-PA75-3
46	50						SD	MS		−20.97	7.53	15.1	40.0	3.1			
47	70						SD	M		−20.52	10.69	14.6	38.5	3.1			
48	38						SD	M		−21.05	10.44	15.5	41.2	3.1			
49	1				75.4		SD	L	L	−20.17	9.02	31.4	81.5	3.0	625–506	523 ± 32	FTMC-PA75-1
50	2				73.3	38.8	SD	L	L	−20.60	9.03	15.5	40.7	3.1			
51	4				39.5	15.7	SD	M	M	−20.89	9.74	16.2	42.1	3.0			
52	78				44.3	14.7	SD	ML	M	−20.82	9.22	15.7	40.7	3.0			
53	80						SM	MS		−21.18	10.09	15.9	40.8	3.0			
54	39						SM	MS		−20.48	10.16	16.2	42.8	3.1			
55	37						SM	M		−20.53	11.00	15.4	40.7	3.1			
56	12				27.6	4.0	UM	S	S	−20.87	10.43	15.4	40.6	3.1	786–686	839 ± 28	FTMC-PA75-6
57	83	Vilnius Lower castle	Royal castle	16th–17th c.						−21.11	10.60	16.2	41.4	3.0			
58	11				74.3	70.0	D	L	L	−20.61	10.45	14.4	37.5	3.1	467–310	327 ± 29	FTMC-PA75-5
59	14						SD	L		−20.54	9.44	15.3	36.6	2.8	305–…	200 ± 28	FTMC-PA75-7
60	82						SM	ML		−20.62	10.27	15.4	39.6	3.0			
61	15				46.3	23.0	SM	M		−21.02	11.64	14.8	37.7	3.0	305–…	207 ± 28	FTMC-PA75-8
62	33						SM	M		−21.36	10.23	16.3	41.6	3.0			
63	81						SM	M		−20.29	10.68	15.4	40.4	3.0			
64	59	Vilnius, Didžioji st. 32	Large town, capital of the Grand Duchy of Lithuania	16th–18th c.	68.1					−20.50	12.67	16.2	45.2	3.3			
65	68	Vilnius, Klaipėdos st. 7a,b								−20.97	11.79	15.6	43.6	3.3			
66	26	Vilnius, Klaipėdos st. 7a,b			63.0	26.0			L	−21.60	10.77	15.8	43.9	3.2			
67	65	Vilnius, Klaipėdos st. 7a,b					SD	M		−20.86	10.78	16.1	44.9	3.3			
68	25	Vilnius, Klaipėdos st. 7a,b			57.0	22.0	SM	ML	ML	−20.45	10.98	15.9	44.0	3.2			
69	64	Vilnius, Radvilų st. 5			64.2	46.8		L	L	−19.94	12.90	14.2	39.5	3.2			
70	29	Vilnius, Radvilų st. 5			55.0	30.0		ML	L	−20.15	11.99	16.2	45.4	3.3			
71	66	Vilnius, Reformatai sq.			39.1	20.2			MS	−21.19	11.09	15.2	42.9	3.3			
72	27	Vilnius, Reformatai sq.					SD	ML		−20.26	10.42	16.3	45.2	3.2			
73	62	Vilnius, Reformatai sq.					SM	M		−20.39	11.77	16.0	44.8	3.3			
74	28	Vilnius, Reformatai sq.					SM	L	L	−19.94	11.59	15.9	41.4	3.0			
75	74	Vilnius, Žygimantų st. 14				4.4			S	−20.59	10.47	15.8	41.2	3.0			

**Table 3 animals-14-01023-t003:** Summary statistics for the various groups of dogs and other species. Species of the wild game: *Bisons bonasus* MNI 43; *Alces alces* MNI 3 *Capreolus capreolus* MNI 1, *Sus scrofa* MNI 1, *Lepus timidus/L. Europaeus* MNI 2.

Site	Dating, AD	MNI	δ^13^C, ‰					δ^15^N, ‰					References
			Mean	SD	Min	Max	Range	Mean	SD	Min	Max	Range	
Masteikiai cemetery	12th–13th c.	4	−20.2	0.3	−20.6	−19.9	−0.8	9.5	1.2	8.3	10.8	2.5	This study
Kernavė medieval town and hillfort	13th–14th c.	9	−20.9	0.3	−21.4	−20.2	−1.1	9.2	0.9	7.8	10.8	3.0	This study
Klaipėda (Memelburg) Castle, coastal	late 13th–early 14th c.	5	−21.2	0.7	−21.9	−20.0	−1.8	11.6	1.1	9.9	12.7	2.8	This study
Vilniaus Lower Castle	13th–15th c.	27	−20.7	0.3	−21.3	−20.0	−1.2	9.8	1.2	7.5	13.1	5.5	This study
Vilniaus Lower Castle	16th–17th c.	7	−20.8	0.4	−21.4	−20.3	−1.1	10.5	0.7	9.4	11.6	2.2	This study
Vilnius town	16th–18th c.	12	−20.6	0.5	−21.6	−19.9	−1.7	11.4	0.8	10.4	12.9	2.5	This study
Kaunas town	16th–18th c.	5	−20.5	0.2	−20.8	−20.2	−0.6	11.3	1.5	8.8	12.9	4.1	This study
Memelburg (Klaipėda) town, coastal	16th–17th c.	2	−20.5	0.1	−20.5	−20.4	−0.1	9.7	0.3	9.5	9.9	0.4	This study
Palanga, small town, coastal	17th–18th c.	1	−20.6	-				11.3				-	This study
Small towns	16th–18th c.	3	−20.7	0.3	−21.0	−20.4	−0.6	10.7	1.1	9.7	12.1	2.5	This study
**Dogs, total**	**12th–18th c.**	**75**	**−20.7**	**0.4**	**−21.9**	**−19.9**	**−2.0**	**10.3**	**1.3**	**7.5**	**13.1**	**5.6**	
Humans, Vilnius Elite	16th–18th c.	47	−19.9	0.3	−20.4	−19.2	−1.2	11.6	0.7	9.5	13.5	4.0	[53]
Humans, Vilnius commoners	16th–18th c.	39	−20.0	0.4	−20.5	−19.1	−1.4	10.9	0.8	10.1	12.2	2.1	[53]
Humans, Lithuania rural, inland	15th–18th c.	83	−20.3	0.3	−21.2	−19.5	−1.6	10.2	0.8	8.5	12.2	3.6	[53,54]
Pigs, Lithuania	15th–18th c.	19	−21.7	0.6	−22.8	−20.8	−2.0	7.9	1.2	5.6	10.0	4.4	[21,22]
Cattle, Litrhuania	15th–18th c.	9	−22.0	0.8	−22.8	−21.2	−1.6	6.7	2.2	4.5	8.9	4.4	[21,22]
Wild game, Lithuania	15th–18th c.	50	−22.9	0.7	−24.6	−20.8	−3.8	4.7	1.0	1.9	6.2	4.3	[21,22,46]

## Data Availability

Data are contained within the article and Appendix A.

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
