# Peer review of "Dogs in Lithuania from the 12th to 18th C AD: Diet and Health According to Stable Isotope, Zooarchaeological, and Historical Data"

_animals, 2024, doi:10.3390/ani14071023_

Round 1

Reviewer 1 Report

Comments and Suggestions for Authors

This paper describes isotopic and, very briefly a morphological analysis of dogs from Lithuania spanning the 12th to 18th centuries. Overall, the paper is very good with interesting results. I do not see any supplementary information and this would be a great opportunity to publish more details about the dog remains. I see that the authors refer to their other work on the morphology of the dogs. However, one is in press and the other paper is not in English so I cannot judge if the relevant data on the metrics is published. It would be good so have the metric data for the individual dogs and all of their skeletal elements published in supplementary material as well as the box plots of the reconstructed withers heights. Additionally, it the supplementary information could contain a fuller description of the pathologies of the dogs so that other researchers can use the data.

Overall, the paper is very useful, but I would encourage the authors to provide more details of the remains in supplementary information. The historical background and information given about the diet of the canines is very informative and makes a very useful addition to the publication.

Minor comments

The description of the samples could be simplified as they are referred to in several different sections. Reorganization to have all the data in one section then tables with NISP and MNI count would be helpful for the reader.

Add line number throughout to help reviewers

Some of the references are in red and on page 3 the text “Due to unforeseen events, we lost contact with that new academic editor, and we would like to ask your thoughts on this manuscript again. - what do you think about this?” does not make sense. Are these notes that were not deleted prior to submission.

Page 8 third paragraph buit should be built

Comments on the Quality of English Language

The English is very good, only minor corrections and re-reading of the paper prior to publication.

Author Response

We are very grateful to the reviewer for his very accurate comments and good advises. See our responses below, please.

This paper describes isotopic and, very briefly a morphological analysis of dogs from Lithuania spanning the 12th to 18th centuries. Overall, the paper is very good with interesting results. I do not see any supplementary information and this would be a great opportunity to publish more details about the dog remains. I see that the authors refer to their other work on the morphology of the dogs. However, one is in press and the other paper is not in English so I cannot judge if the relevant data on the metrics is published. It would be good so have the metric data for the individual dogs and all of their skeletal elements published in supplementary material as well as the box plots of the reconstructed withers heights. Additionally, it the supplementary information could contain a fuller description of the pathologies of the dogs so that other researchers can use the data.

Overall, the paper is very useful, but I would encourage the authors to provide more details of the remains in supplementary information.

Supplementary data with dog measurements and other details added.

Minor comments

The description of the samples could be simplified as they are referred to in several different sections. Reorganization to have all the data in one section then tables with NISP and MNI count would be helpful for the reader.

Corrected

Add line number throughout to help reviewers.

Sorry for this, done...

Some of the references are in red and on page 3 the text “Due to unforeseen events, we lost contact with that new academic editor, and we would like to ask your thoughts on this manuscript again. - what do you think about this?” does not make sense. Are these notes that were not deleted prior to submission.

sure...

Page 8 third paragraph buit should be built

Corrected

Reviewer 2 Report

Comments and Suggestions for Authors

This paper presents the results of stable carbon and nitrogen analysis of dog bone from 13th-18th century Lithuania. Bone from 75 dogs from several sites spanning this time were analyzed. The authors provide dietary interpretations of the isotope ratios taking into account contexts of the dog remains and physical attributes of the dogs.

Overall, I find the interpretation less than convincing. It seems that the contexts of the dog remains factors too strongly into the interpretations of the isotope ratios. At best, the interpretations are simply suppositions, and I cannot state that the data support the conclusions.

I strongly encourage the authors to preform Bayesian dietary mixing model analyses of the isotope data. You have isotope data from various animal sources, can possibly obtain data on plants and feces from the literature. Doing so will allow you to be less subjective in your interpretations while admitting the limitations of the models.

Beyond that general issue, I have comments on specific issues in the text.

1. The second sentence of the introduction is misleading. The vast majority of contemporary dogs are free roaming. This is well established in the literature. So the sentence needs to be revised or deleted.

2. The first three sentences of the third paragraph starting on page 3 needs references.

3. Throughout the paper you refer to dog "types". In one instance you mention "morphotypes". If this is what you mean by "types" it needs to be made clear. If you mean something else, you need to make that clear.

4. The last sentence of the first paragraph under 2.1. δ13C and δ15N stable isotopic analysis needs to be deleted.

5. On page 4, more information is needed for the radiocarbon dates. The lab(s) need to be specified. Are these AMS dates? Have they been published previously? If so, provide a citation. If not, you need to follow standard procedures for reporting new dates. Given that you interprate the isotope ratios to indicate freshwater/brackish water fish and possibly marine resources, you need to comment on the possibility of carbon reservoir effects on the dates. Do you anticipate that there are offsets?

6. The 2.3 Site description subsection needs to be moved ahead of the Materials and methods section as a section and expanded to include information on the specific find locations of the dog remains for each site. Were the bones found in pit features, middens, or general contexts within stratigraphy. This can potentially be done in a table or in a supplemental data file. It would seem that the specific contexts of the finds would have some implication for the treatment of the dogs in life. You should also indicate where the bones are curated along with any catalog/accession numbers.

7. On line 28 of page 16, where are the isotope data for feces?

8. On Figure 6, minus signs need to be added to the medians.

9. On line 48 on page 17, "lowest" is preferable to "most depleted".

10. The sentence on line 89 on page 19 needs a citation.

Comments on the Quality of English Language

Some of the English is a bit awkward and can use editing.

Author Response

Thanks the reviewer for the very valuable comments and suggestions. We hope that corrections we made have improved the manuscript.

The authors provide dietary interpretations of the isotope ratios taking into account contexts of the dog remains and physical attributes of the dogs. Overall, I find the interpretation less than convincing. It seems that the contexts of the dog remains factors too strongly into the interpretations of the isotope ratios. At best, the interpretations are simply suppositions, and I cannot state that the data support the conclusions.

I strongly encourage the authors to preform Bayesian dietary mixing model analyses of the isotope data. You have isotope data from various animal sources, can possibly obtain data on plants and feces from the literature. Doing so will allow you to be less subjective in your interpretations while admitting the limitations of the models.

The reconstruction of dog diets and the analysis of stable isotope data presented challenges of interpretation. The main problem was the equivalence of the isotopic values, which can be caused by different factors. We tried to explain the results with very different options.  And yes, certainly various other sources - historical data, the condition of the dogs' teeth, etc. - were very important when interpreting stable isotope results. These data were the basis for selecting the most probable scenario. We also made Bayesian dietary mixing modelling for reconstructing dietary patterns. However modelling the diets of omnivores that lived in very different environments and under different conditions, which are only predictable to the researcher, leaves many possible scenarios and many unknown options. The available food source data, archaeological, zooarchaeological contexts and historical data, as well as priors selected by researcher have a strong impact on the modelling results. Modelling should therefore be treated with caution, and in the case of the material analysed in this study, the other sources of a different nature - biometry, health status, historical records played the key role in both hypothesis development and testing, as well as in supporting the results of δ13C and δ15N stable isotope analysis.

Beyond that general issue, I have comments on specific issues in the text.

  1. The second sentence of the introduction is misleading. The vast majority of contemporary dogs are free roaming. This is well established in the literature. So the sentence needs to be revised or deleted.

Sentence corrected

2. The first three sentences of the third paragraph starting on page 3 needs references.

I'm not sure if we found this place. We checked the text and added references in a few places

3. Throughout the paper you refer to dog "types". In one instance you mention "morphotypes". If this is what you mean by "types" it needs to be made clear. If you mean something else, you need to make that clear.

Corrected, only "type" left

4. The last sentence of the first paragraph under 2.1. δ13C and δ15N stable isotopic analysis needs to be deleted.

Sentence corrected

5. On page 4, more information is needed for the radiocarbon dates. The lab(s) need to be specified.

Lab already was specified in the first text (previous lines 156-157). Now information is provided (lines 306-322)

Are these AMS dates?

Yes, information added

Have they been published previously? If so, provide a citation. If not, you need to follow standard procedures for reporting new dates.

Dates are publishing in this paper (specified, lines 301-302). Dates are reporting following the standard procedures (see lines 306-322, Table 2)

Given that you interprate the isotope ratios to indicate freshwater/brackish water fish and possibly marine resources, you need to comment on the possibility of carbon reservoir effects on the dates. Do you anticipate that there are offsets?

A few individuals from coastal sites, for which fish might play a more important role in the diet, were not AMS 14C dated. For the analysed inland dogs, fish was an insignificant food source. Therefore, the reservoir effect should not have affected AMS 14C dates. Moreover, all the AMS 14C dates we got, correspond to the dating of the dogs according to the archaeological contexts.

6. The 2.3 Site description subsection needs to be moved ahead of the Materials and methods section as a section and expanded to include information on the specific find locations of the dog remains for each site. Were the bones found in pit features, middens, or general contexts within stratigraphy. This can potentially be done in a table or in a supplemental data file. It would seem that the specific contexts of the finds would have some implication for the treatment of the dogs in life. You should also indicate where the bones are curated along with any catalog/accession numbers.

The structure of the manuscript was reorganized.  Information on the specific find locations of the dog remains was already provided in the section "site description" -  all dog remains (except of Masteikiai cemetery) were found in the cultural layers of the sites. Information where dog remains are stored also was provided (section "Biometrical analysis"). For this manuscript version, Supplement tables were created and dog catalog number (when available) was included in the tables with the dog bones measurements (S1). However, for every individual in this study ID number was given, and it also works as catalog number.

7. On line 28 of page 16, where are the isotope data for feces?

We included feces values for the Bayesian modelling (see Bayesian modelling section, please)

8. On Figure 6, minus signs need to be added to the medians.

Corrected

9. On line 48 on page 17, "lowest" is preferable to "most depleted".

Changed

10. The sentence on line 89 on page 19 needs a citation.

Added

Reviewer 3 Report

Comments and Suggestions for Authors

This study presents the results of stable carbon and nitrogen isotope analyses of bone collagen from a large collection of dogs (N=75), found at various archaeological sites in Lithuania dating from the 12th to the 18th centuries AD. This study corpus is a subsample of a larger corpus on which biometrics and health have been assessed and previously published. Results in terms of diet are compared over time (medieval/modern period), as a function of the socio-economic status (elite/non elite), location (coastal/inland), biometry (dog size on a site scale), and with human diet (site scale). The results show (1) lower d15N values in the early modern than in the medieval period, in line with the contribution of wild game to the diet of elite hunting dogs in the medieval period and the declining importance of hunting in the early modern period; (2) that height at withers and diet are not correlated; (3) that dogs consumed more fish at coastal sites than inland; (4) great variability in the diet of dogs in elite contexts, perhaps reflecting the fact -attested form historical records - that dogs could travel as gifts.

Overall, the strengths of the study are the great size of the corpus, the variety of contexts studied, a good knowledge of the context and historical sources documenting dog feeding and training, and the possible combination of the results of stable isotope analysis with biometric data and data relating to the health status of these dogs. However, the large number of variables in a limited number of contexts, and a disparity in sample sizes, sometimes weakens the validity of interpretations derived from comparisons. The results are really interesting but they could be reinforced by a major reorganization of the text to provide greater clarity on the following points in particular:

(1) the data relating to biometry and the dogs health status have been published earlier (bibliographic references 15 &16, as clearly stated in the introduction and recalled later in the 3.1 section). As a result, the text lacks clarity on the extent to which the conclusions relating to these specific parts of the study and reported in the present manuscript are new or constitute syntheses of what has been published elsewhere. It seems to me that the latter is correct for biometrics, but I cannot be sure about the health data. In my opinion, it is not a problem at all to reuse these data in this present study, but depending on the answer, their descriptions (currently placed in the Methods and Results sections) might be better placed in a "Background" section immediately following the introduction, or in the discussion;

(2) Placing the conclusions of biometric and health studies in a "background" section would also make it possible to set out working hypotheses from the beginning. For a major problem for interpretation is the equifinality of the isotopic values interpreted here. This problem is not inherent to this particular study, it is inherent to the nature of the dataset. The authors themselves are aware of this problem, and propose a series of possible interpretations at various points in the manuscript (for example: page 16 lines 5-16: higher d15N values in dog bone collagen could be due to a higher proportion of meat or fish in the diet; or starvation; or “on the contrary” a decreasing proportion of wild game in the diet). The problem remains: none of the proposed interpretations can be validated a posteriori rather than another. The only way out of this impasse is to propose hypotheses supported by sources of a different nature (biometry, health status, historical sources), which will then be tested by isotope analyses.

(3) The results and discussion sections are currently merged, and should be clearly distinguished.

Other comments:

Introduction

The introduction is well organized and informative but could be more synthetic. In particular, the part describing results of previous stable isotope analyses in bone collagen from dogs dating from the Neolithic to Bronze Age (page 2, lines 22-45) should be shortened (no need to give the details of stable isotope values, for example).

Page 2 Lines 37-39 “These values demonstrate that the major dietary input was plant-based food, with the addition of some meat; this diet was also typical for the medieval dogs presented in this study » the results from the present study should not be given in the introduction.

Page 3 lines 14-15 :” This paper focused on the nutrition and related health issues of the dogs”: the results relating to the dogs' health are not really compared with those relating to their diet. In fact the data relating to age and health are placed in the very last section of the manuscript just before the conclusion and their description makes no reference to the results of isotope analysis. It is true though, that earlier in the manuscript (section 3.2, page 16-17 lines 26-) there is an attempt to link both datsets ( “The relatively high δ15N values of the urban canines from Vilnius and Kaunas were probably not due to higher meat or fish consumption, but rather to the high proportion of faeces and animal waste (i.e., bones) of city dwellers comprising their diet, as well as chronic starvation. Presumably, a high number of the urban canines were stray dogs, suffering from food shortages and feeding on faeces and bones (for zooarchaeological data, see below)”: the fact that this discussion refers to data presented later in the text again points to a problem with the text's organization.

Page 3 last lines in the introduction: “to look at how similar the diets of dogs and pigs, i.e., an-imals with similar lifestyles, might be, at least in urban areas.” The similarity in pigs and dogs’ lifestyles is a problematic presupposition, given that this is precisely what will be tested by comparing their isotopic values (pigs can also be herded extensively). So there is a bit of a contradiction here.

Material and methods

Page 3 last lines “Due to unforeseen events, we lost contact with that new academic editor, and we would like to ask your thoughts on this manuscript again. - what do you think about this? » must be removed !

Page 4 lines 10 & 11: “Repeated analyses of laboratory working standards gave standard deviations of less than ±0.1‰ for carbon and ±0,15‰ for nitrogen”: replace carbon and nitrogen by “d13C values” and “d15N values”

Page 4 line 15 “collagen quality parameters were calculated, the main of them is C/N atomic ratio, which fell in the range of 2.8 -3.6 as suggested by many »: the correct lower value is 2.9 instead of 2.8 (please check agin in DeNiro, 1985). According to this and looking at the data given in Tale 1, two of the collagen extracts reported in the study should be disqualified: dog ID 77 and dog ID 14.

Table 1: the %N and %C should be reported to one decimal place (see Guiry and Szpak, 2021).

Page 6 site description: for the readers not familiar with Lithuania, it is not clear, which group the Masteikiai cemetery refers to (rural or urban population?)

Results and discussion

The 3.1 section is very interesting, but should definitely be placed in a “background” section.

Page 17 line 51 “ the slightly lower carbon values of the Masteikiai dogs may reflect the presence of millet in their diet” replace by “the lightly HIGHER carbon stable isotope values”

There is no need to mention the contribution of C4 plants to diet for d13C values around -20‰, unless there is clear evidence (from archaeobotanical data for example) for millet consumption at the site.

Page 19 lines 106-108: “Meanwhile, the nutrition of the other largest dogs, sighthound-type individuals ID 1 and ID 2, or the large Molossian-type canines ID 9 and ID 11, did not differ (p > 0.05) from that of the other dogs » unless the measured signals fail showing a difference. What is shown is no difference in the stable isotope composition, not in the nutrition.

Page 20 lines 135-136: “A large number of fish remains have been found in Klaipeda/Memelburg castle, of which freshwater fish predominated (99.%)”: are higher or lower d13C values expected for freshwater fish in this context?

Lines 149-151: “The carbon isotopic value (…) was lower than those of the inland canines. The carbon and nitrogen isotopic values compared to the other Klaipeda dogs would suggest a lower presence of fish and meat in the diet of ID 21.”: does this presuppose higher d13C values in freshwater fish in the area? Can this be argued?

Page 21 Diet of elite dogs: Historical records. It is not clear at all, how this information relates to the results of the present study.

Figure 9: for a non-specialist like me, it is not clear, what the arrows next to the teeth indicate. I assume it is alveolar inflammation, but I can't see anything in these photographs.

Conclusion

Page 25 lines 274-277: “The results of our study confirm that the size, morphotype and health of canines from different time periods and sociocultural environments varied. This reflects the different living conditions, care and nutrition of the dogs, as well as the diversity of their roles and functions in the daily lives of humans.” Does this relate to the results from the present study, or an earlier study?

Lines 279-281 “The stable isotopic analysis supported the historical evidence, indicating that cereals were highly important in the diet of elite dogs. » I could not find a demonstration of this in the text.

Author Response

We would like to thank the reviewer very much for his very valuable comments and careful work. Here's our response:

(1) the data relating to biometry and the dogs health status have been published earlier (bibliographic references 15 &16, as clearly stated in the introduction and recalled later in the 3.1 section). As a result, the text lacks clarity on the extent to which the conclusions relating to these specific parts of the study and reported in the present manuscript are new or constitute syntheses of what has been published elsewhere. It seems to me that the latter is correct for biometrics, but I cannot be sure about the health data. In my opinion, it is not a problem at all to reuse these data in this present study, but depending on the answer, their descriptions (currently placed in the Methods and Results sections) might be better placed in a "Background" section immediately following the introduction, or in the discussion;

Thank you very much for this brilliant advice. We have seen this problem as well, but have not come up with a solution.  We corrected it according to this comment, and made "Background" section

(2) Placing the conclusions of biometric and health studies in a "background" section would also make it possible to set out working hypotheses from the beginning. For a major problem for interpretation is the equifinality of the isotopic values interpreted here. This problem is not inherent to this particular study, it is inherent to the nature of the dataset. The authors themselves are aware of this problem, and propose a series of possible interpretations at various points in the manuscript (for example: page 16 lines 5-16: higher d15N values in dog bone collagen could be due to a higher proportion of meat or fish in the diet; or starvation; or “on the contrary” a decreasing proportion of wild game in the diet). The problem remains: none of the proposed interpretations can be validated a posteriori rather than another. The only way out of this impasse is to propose hypotheses supported by sources of a different nature (biometry, health status, historical sources), which will then be tested by isotope analyses.

Corrected according to the suggestion.

(3) The results and discussion sections are currently merged, and should be clearly distinguished.

We chose this format and decided to merge the sections to avoid overlapping and repetition in the text and to make the text easier to read.  As we did not receive any comments on this issue from other reviewers, we have decided to keep these chapters merged. 

Other comments:

Introduction

The introduction is well organized and informative but could be more synthetic. In particular, the part describing results of previous stable isotope analyses in bone collagen from dogs dating from the Neolithic to Bronze Age (page 2, lines 22-45) should be shortened (no need to give the details of stable isotope values, for example).

Corrected, shortened

Page 2 Lines 37-39 “These values demonstrate that the major dietary input was plant-based food, with the addition of some meat; this diet was also typical for the medieval dogs presented in this study » the results from the present study should not be given in the introduction.

Corrected, removed.

Page 3 lines 14-15 :” This paper focused on the nutrition and related health issues of the dogs”: the results relating to the dogs' health are not really compared with those relating to their diet. In fact the data relating to age and health are placed in the very last section of the manuscript just before the conclusion and their description makes no reference to the results of isotope analysis. It is true though, that earlier in the manuscript (section 3.2, page 16-17 lines 26-) there is an attempt to link both datsets ( “The relatively high δ15N values of the urban canines from Vilnius and Kaunas were probably not due to higher meat or fish consumption, but rather to the high proportion of faeces and animal waste (i.e., bones) of city dwellers comprising their diet, as well as chronic starvation. Presumably, a high number of the urban canines were stray dogs, suffering from food shortages and feeding on faeces and bones (for zooarchaeological data, see below)”: the fact that this discussion refers to data presented later in the text again points to a problem with the text's organization.

Corrected. We reorganized the text and also linked health and nutrition issues more clearly

Page 3 last lines in the introduction: “to look at how similar the diets of dogs and pigs, i.e., an-imals with similar lifestyles, might be, at least in urban areas.” The similarity in pigs and dogs’ lifestyles is a problematic presupposition, given that this is precisely what will be tested by comparing their isotopic values (pigs can also be herded extensively). So there is a bit of a contradiction here.

We corrected the sentence.

Material and methods

Page 3 last lines “Due to unforeseen events, we lost contact with that new academic editor, and we would like to ask your thoughts on this manuscript again. - what do you think about this? » must be removed !

Removed

Page 4 lines 10 & 11: “Repeated analyses of laboratory working standards gave standard deviations of less than ±0.1‰ for carbon and ±0,15‰ for nitrogen”: replace carbon and nitrogen by “d13C values” and “d15N values”

Replaced

Page 4 line 15 “collagen quality parameters were calculated, the main of them is C/N atomic ratio, which fell in the range of 2.8 -3.6 as suggested by many »: the correct lower value is 2.9 instead of 2.8 (please check again in DeNiro, 1985). According to this and looking at the data given in Tale 1, two of the collagen extracts reported in the study should be disqualified: dog ID 77 and dog ID 14.

We have selected to use C:N ratio range of 2.8–3.6  as it is used in a number of studies, both recent and previous, e.g.:

Ambrose, Stanley H. "Preparation and characterization of bone and tooth collagen for isotopic analysis." Journal of archaeological science 17.4 (1990): 431-451.

Le Huray, Jonathan D., Holger Schutkowski, and Michael P. Richards. "La Tène dietary variation in Central Europe: a stable isotope study of human skeletal remains from Bohemia." The social archaeology of funerary remains. Oxford: Oxbow Books (2006): 99-121.

Pate, F. Donald, Renata J. Henneberg, and Maciej Henneberg "Stable carbon and nitrogen isotope evidence for dietary variability at ancient Pompeii, Italy." Mediterranean Archaeology & Archaeometry 16.1 (2016)

Madden, Odile, et al. "Quantifying collagen quality in archaeological bone: Improving data accuracy with benchtop and handheld Raman spectrometers." Journal of Archaeological Science: Reports 18 (2018): 596-605.

Cleland, Timothy P., Julianne J. Sarancha, and Christine AM France. "Proteomic profile of bone “collagen” extracted for stable isotopes: Implications for bulk and single amino acid analyses." Rapid Communications in Mass Spectrometry 35.6 (2021): e9025.

France, Christine AM, and Douglas W. Owsley. "Stable carbon and oxygen isotope spacing between bone and tooth collagen and hydroxyapatite in human archaeological remains." International Journal of Osteoarchaeology 25.3 (2015): 299-312.

Wang, Fen, Chao Yuan, and Shiling Yuan. "Radiocarbon dating and diet: the Jiaojia site in China." Radiocarbon 62.5 (2020): 1515-1523.

Table 1: the %N and %C should be reported to one decimal place (see Guiry and Szpak, 2021).

Corrected to one decimal place

Page 6 site description: for the readers not familiar with Lithuania, it is not clear, which group the Masteikiai cemetery refers to (rural or urban population?)

Corrected, explained

Results and discussion

The 3.1 section is very interesting, but should definitely be placed in a “background” section.

"Background" section created and 3.1 section replaced to it.

Page 17 line 51 “ the slightly lower carbon values of the Masteikiai dogs may reflect the presence of millet in their diet” replace by “the lightly HIGHER carbon stable isotope values”

Corrected

There is no need to mention the contribution of C4 plants to diet for d13C values around -20‰, unless there is clear evidence (from archaeobotanical data for example) for millet consumption at the site.

Masteikiai cemetery is a burial site, excavations were carried out here 30 years ago, and no archaeobotanical research or sampling was done. However, there is presence of millet in other 12-14th C AD sites close by, e.g.  in Kernavė, 50 km away. This suggested a possible consumption of millet by the community buried in Masteikiai

Page 19 lines 106-108: “Meanwhile, the nutrition of the other largest dogs, sighthound-type individuals ID 1 and ID 2, or the large Molossian-type canines ID 9 and ID 11, did not differ (p > 0.05) from that of the other dogs » unless the measured signals fail showing a difference. What is shown is no difference in the stable isotope composition, not in the nutrition.

Corrected

Page 20 lines 135-136: “A large number of fish remains have been found in Klaipeda/Memelburg castle, of which freshwater fish predominated (99.%)”: are higher or lower d13C values expected for freshwater fish in this context?

Corrected, explained (lower carbon values).

Lines 149-151: “The carbon isotopic value (…) was lower than those of the inland canines. The carbon and nitrogen isotopic values compared to the other Klaipeda dogs would suggest a lower presence of fish and meat in the diet of ID 21.”: does this presuppose higher d13C values in freshwater fish in the area? Can this be argued?

Text corrected, d13C values of fish in the area were not higher. Values of fish included to Figure 11.

Page 21 Diet of elite dogs: Historical records. It is not clear at all, how this information relates to the results of the present study.

We have separated this section and not combined it with, e.g., the section of the isotope results, in order to make the text more clear. We corrected it by making it more linked to the results of stable isotope analysis and we would like to keep it separated.

Figure 9: for a non-specialist like me, it is not clear, what the arrows next to the teeth indicate. I assume it is alveolar inflammation, but I can't see anything in these photographs.

Figure 9 corrected. Agree, it was not clear, shame on me...

Conclusion

Page 25 lines 274-277: “The results of our study confirm that the size, morphotype and health of canines from different time periods and sociocultural environments varied. This reflects the different living conditions, care and nutrition of the dogs, as well as the diversity of their roles and functions in the daily lives of humans.” Does this relate to the results from the present study, or an earlier study?

Corrected, in general removed. Agree, it is related to the another study.

Lines 279-281 “The stable isotopic analysis supported the historical evidence, indicating that cereals were highly important in the diet of elite dogs. » I could not find a demonstration of this in the text.

Corrected. We have improved the demonstration and link between the stable isotope and historical data in a few chapters ( Dog nutrition in the 12th to 18th C AD: Stable isotope results and Diet of elite dogs: Historical records)

Round 2

Reviewer 1 Report

Comments and Suggestions for Authors

The authors have amended the paper following the recommendations I made in the initial review

Reviewer 2 Report

Comments and Suggestions for Authors

The revised paper is much improved from the original.

Comments on the Quality of English Language

The quality of the English is acceptable.